# Measuring Language Model Uncertainty with Internal Concepts

## Abstract

We study the problem of evaluating the predictive uncertainty of large language models (LLMs). We assign an uncertainty measure to the correctness of outputs from an LLM conditioned on a query using a form of entropy that applies to semantic objects (concepts). Unlike prior works, the notion of meaning used to define concepts is derived from the LLM, rather than from an external model. Our method measures an uncertainty over concept structures by drawing from ideas in Formal Concept Analysis (FCA) and lattice/order theory, and can be used to estimate correctness in closed- and open-ended scenarios. Our method has a relative improvement of up to 4.8% on average across five standard benchmarks as well as improves over comparable baselines on datasets consisting of both closed- and open-ended questions.

## 1 Introduction

Large language models (LLMs) produce factually incorrect output, colloquially referred to as *hallucination*, which is a serious concern for their widespread use. For instance, LLM hallucination has already been the center of multiple scandals such as naming academics as terrorists (Hsu, 2023), inaccurately summarizing a federal court case using non-existent legal precedence (Belanger, 2023), and citing imaginary scientific articles (Heaven, 2022).

Recent work shows that hallucinations often occur when an LLM has high *uncertainty* (Farquhar et al., 2024). However, measuring a robust notion of LLM uncertainty for detecting hallucination is challenging (Abbasi-Yadkori et al., 2024; Hou et al., 2024). This difficulty stems from the fact that the LLM's output logits reflect *syntactic* uncertainty — the probability of individual tokens — while hallucination is a property of the *meaning* of the output. As a solution to this issue, Kuhn et al. (2023) proposed clustering sampled outputs based on their meaning before computing an entropy measure (dubbed 'semantic entropy'). In their approach, meaning is defined using an external model which serves as an arbiter of semantic equivalence.

Semantic entropy improves over naive entropy, but semantic variation of an LLM's output does not always align with its uncertainty. For instance, as noted by Abbasi-Yadkori et al. (2024), semantic entropy becomes large in open-ended scenarios even when an LLM responds correctly. These scenarios are common and can arise together with closed-ended questions. For instance, the query *"Name a city in the 31st US state"* introduces uncertainty on two levels: first, determining which state is the 31st — a question with a definite answer — and second, identifying an appropriate city within that state — a question that is inherently open-ended.

Another challenge comes from defining semantic equivalence. Existing methods often use an external model of semantics, but a semantic grouping which differs from a model's own notion of semantics incorporates external (lack of) knowledge into the uncertainty measure. For instance, the model used by Kuhn et al. (2023) designates *"Kitzbuehel"* and *"Kitzbuehel, Austria"* as distinct even though a relatively small LLM correctly outputs both as answers to *"Where does the infamous 'Streif' downhill ski race take place?"*. In addition, the external model limits the applicable domains to its training distribution which is often much smaller than the LLM's training distribution.

We address these problems with Conceptual Uncertainty, a framework for measuring LLM uncertainty, which applies to both open- and closed-ended scenarios without using external knowledge. Our key insight is to measure uncertainty over an LLM's likelihood of different answer truth as-

Figure 1: A walkthrough of Conceptual Uncertainty. After sampling answers to the question $q$, we construct a matrix of the likelihoods that each premise implies each answer. As shown on the right of step 1, this context encodes the concept structure, or sets of consistent answers and each set has an associated likelihood. In step 2, we leverage the structure from step 1 to compute the probability of all truth assignments to all answers, and finally we use Conceptual Entropy to compute a variant of entropy over the truth-assignment distribution from step 3.

signments (a complete assignment of true/false to all answers), defined by the structure of *internal concepts*. "Internal" indicates that these concepts are defined using only the LLM's distribution, removing the dependence on an external model, and the distribution over possible truth assignments is naturally agnostic to the open- and closed-endedness of a question.

Concretely, we define concepts as subsets of candidate answers that are closed under forward implication (*i.e.*, for answers $a$ and $b$, if $a$ is an answer to query $q$, then so is $b$). We propose to extract such implications between answers $a$ and $b$ directly from the LLM by evaluating the likelihood ratio of strings of the form $qbqa$ and $qa$ which can be interpreted as a pointwise mutual information between the premise $qb$ and answer $a$ conditioned on $q$. As we discuss, this is related to *Formal Concept Analysis* (FCA) (Ganter et al., 1997), a mathematical framework for computing a concept hierarchy from general pairings between two sets. From concepts, we define the "truth-assignment likelihood" function, which is a map $P : 2^A \to [0,1]$ that describes the likelihood of various truth assignments (*e.g.*, the likelihood that certain answers are true while others are false). Importantly, the function $P$ depends on the query $q$, and is agnostic to whether $q$ was open- or closed-ended.

Finally, a modified entropy variant is required to measure model uncertainty. Conventional entropy becomes zero when likelihood is concentrated on the "all false" assignment (*i.e.*, no answers are correct). To address this, we propose a measure termed Conceptual Entropy that focuses only on truth assignments that are not all false, and also satisfies a weighted composition property for mutually exclusive concepts. Moreover, we show that it coincides with the non-standard normalization approach for entropy that was used without justification in prior work (Aichberger et al., 2024).

Since computing the complete likelihood distribution is computationally expensive, we focus only on the highest-level concepts which partition the set of candidate answers. Our experiments show that using the LLM to compute both the partition and its truth-assignment likelihood function can lead to a more robust measure of LLM uncertainty. Our approach outperforms existing methods on standard benchmarks and synthetic datasets designed to emulate closed- and open-ended queries.

In summary, our main contributions are as follows:

- We propose a framework, Conceptual Uncertainty, for computing LLM uncertainty through an entropy measure on a distribution of internal concepts, which can be obtained from the LLM itself using a framework similar to classical Formal Concept Analysis (Section 3).

- We present Conceptual Entropy, a variant of entropy which is natural for truth assignments of candidate answers and motivates the unconventional normalization used in prior works (Section 3.3).

- We describe an implementation of our framework that focuses on concept partitions. Our experiments show this strategy handles closed- and open-ended questions, as well as questions with short and long answers, two challenging settings for existing approaches (Section 4.2).

- We demonstrate that Conceptual Uncertainty has relative improvement of up to 4.8% on average across five standard datasets for evaluating uncertainty metrics (Table 3) and up to 10.9% relative improvement on mixtures of open- and closed-ended datasets (Table 2).

## 2 A GENERAL FRAMEWORK FOR LLM UNCERTAINTY

In this section, we give some background on existing methods for LLM uncertainty estimation. We write $\mathcal{A}$ for a vocabulary of tokens and $\mathcal{A}^*$ for the set of variable-length strings with tokens in $\mathcal{A}$. A general uncertainty metric is a function which takes an LLM $M$ and a query $q \in \mathcal{A}^*$ and outputs a score $U(M, q) \in \mathbb{R}$. Informally, the metric should provide information on whether $M$ is likely to produce incorrect outputs given the query $q$. We measure an uncertainty metric's usefulness by

$$\Pr\big[U(M, q_1) < U(M, q_2)\big|\, M \text{ answers } q_1 \text{ correctly and } q_2 \text{ incorrectly}\,\big],$$

where the queries $q_1$ and $q_2$ are sampled from an annotated dataset, and a probability closer to 1 represents a stronger uncertainty metric. Indeed, this evaluation corresponds to the ROC-AUC metric commonly used in prior work on LLM uncertainty (Galil et al., 2023) as well as work on metacognition (Hosseini & Ferrell, 1982). Note that, following Lin et al. (2024), we use the term *uncertainty* as opposed to *confidence* as we view uncertainty as independent of the selected output to the query.

### 2.1 COMPONENTS OF LLM UNCERTAINTY

Most existing works quantify LLM uncertainty with two main steps: 1) generating multiple answers to a query, and 2) measuring some form of answer diversity. The model is considered uncertain when answers are diverse. The measure of answer diversity is central to an uncertainty metric and should reflect *semantic* rather than surface-level syntactic differences between the answers. The exception to this description are methods operating on the hidden activations of the model (before producing an answer) to derive an uncertainty score (Kadavath et al., 2022; Yin et al., 2024).

We further decompose existing diversity measures (step 2) into three key components. In the following, we assume we have used a model $M$ and some sampling method to generate $n$ answers $A = \{a_1, \ldots, a_n\} \subset \mathcal{A}^*$ to a query $q \in \mathcal{A}^*$.

**Output Grouping**: The first component is a strategy for grouping the model's answers into (disjoint) subsets $G = \{g_1, \ldots, g_k\}$, $g_i \subset A$, each consisting of semantically equivalent answers. Prior works group using an external entailment model (Kuhn et al., 2023), prompting (Farquhar et al., 2024), or a token-level heuristic (Abbasi-Yadkori et al., 2024) such as the ROUGE (Lin, 2004). Some work also considers a soft grouping with a similarity metric between all answers (Chen et al., 2024).

**Group Weighting**: The second component assigns a likelihood score $P(g_i) \in [0, 1]$ to each of the semantic groups $g_i \in G$. The simplest approach is to sum the model's raw likelihoods for each output in the group, $P(g_i) = \sum_{a \in g_i} M(a|q)$. However, this can be problematic as longer sequences tend to have lower likelihoods. To address this, some methods instead use a length-normalized likelihood $M(a|q)^{1/|a|}$ where $|a|$ is the number of tokens in $a$ (Kuhn et al., 2023; Malinin & Gales, 2021) or a re-weighted likelihood $\Pi_{i=1}^{|a|} M(a_i|a_{i-1} \ldots a_1 q)^{w_i}$ where the $w_i$s are normalized weights representing token semantic importance (Duan et al., 2024). Methods which use soft answer grouping often implicitly assign equal likelihood to each answer.

**Uncertainty Metric**: The third component is a method which takes as input the groups and their likelihood scores and returns an uncertainty score. A common choice is the Shannon entropy $-\sum_{i=1}^{N} p(g_i) \log p(g_i)$ (Malinin & Gales, 2021). The mutual information of all groups is used in recent work (Abbasi-Yadkori et al., 2024). However, group likelihood scores typically do not sum to 1, and existing methods differ in how they handle this issue. Some use unnormalized group likelihoods (Kuhn et al., 2023), normalized likelihoods (Farquhar et al., 2024; Abbasi-Yadkori et al., 2024), or partially normalized likelihoods (Aichberger et al., 2024). Empirically, Aichberger et al. (2024) find that normalizing likelihoods only outside of the log for entropy improves performance.

We summarize the design of existing methods in Table 8 in Appendix F. In Appendix F.1 we present experiments comparing the effectiveness of different choices for each component.

### 2.2 CHALLENGES IN LLM UNCERTAINTY ESTIMATION

We highlight a few challenges that existing methods face that we aim to address in this work:

**External Semantic Model:** Many existing works rely on an external model or method to determine the output grouping, rather than seeking to extract a notion of semantics from the LLM itself. This

---

**Algorithm 1** Conceptual Uncertainty

---

**Input:** A question $q \in \mathcal{A}^*$, LLM $M : \mathcal{A}^* \rightarrow [0, 1]$, number of samples $n$, and distance threshold $\epsilon$.
**Output:** A scalar representing the uncertainty.
 1: $A \leftarrow a_1, a_2, \ldots, a_n \sim M(q)$.      ▷ Sample $n$ answers using multinomial sampling
 2: $r_i \leftarrow [I_q(a_i, a_1), I_q(a_i, a_2), \ldots, I_q(a_i, a_n)]$ for all $i$.      ▷ Construct answer representations
 3: $d_{ij} \leftarrow 1$ if $L_\infty(\log r_i - \log r_j) < \epsilon$ else 0, for all $i, j$.
 4: Compute connected components of adjacency matrix $d$ as $G$ which consists of subsets of $A$.
 5: $P_g(\top) \leftarrow \sum_{a \in g} M(a|q)$ and $P_g(\bot) \leftarrow 1 - P_g(\top)$ for all $g \in G$.
 6: $H_c[P_g] \leftarrow -\log P_g(\top)$ for all $g \in G$.
 7: **return** $\sum_{g \in G} w_i H_c[P_g]$ where $w_i = \frac{1 - P_g(\bot)}{1 - \sum_{g \in G} P_g(\bot)}$.

---

makes the uncertainty not intrinsic to the LLM, and moreover the external model might not have the same knowledge as the LLM (potentially less aligned with human semantics). We show real examples of output grouping failures from using an external model on the left and right of Figure 3.

Some methods do not use an external semantic model, such as that from Chen et al. (2024), but rely on hidden activations to determine output grouping, but such internal representations are not always available and have been shown to poorly represent semantic relations (Liu et al., 2024).

**Closed-Ended vs. Open-Ended Queries:** Most existing approaches cannot handle open-ended queries, where semantically different answers can all constitute valid responses. For example, a correct answer to the query *"Name a city in California"* could be *"Los Angeles," "San Diego," "San Francisco,"* etc., all of which are semantically distinct but equally valid. The middle of Figure 3 shows an example where existing approaches separate equivalently valid answers.

**Semantic Structure:** In certain cases, semantic relations between a set of outputs are not modeled well by a simple partition, but rather should be seen as partially ordered. For example, if a model is prompted with *"Name a type of instrument."*, and returns the outputs *"violin," "stringed instrument,"* and *"Stradivarius violin,"* these outputs have a hierarchical semantic structure that is not well captured by just grouping them into independent clusters.

## 3   Conceptual Uncertainty

In this section, we present our framework for addressing the above challenges. First, we introduce the structure of internal concepts based on entailment orderings. We then define the truth-assignment likelihood function and our proposed Conceptual Entropy measure to measure uncertainty of the truth-assignment distribution. Finally, we describe our practical implementation of these elements.

### 3.1   Internal Concepts

Given a set of candidate answers $A = \{a_1, \ldots, a_n\} \subset \mathcal{A}^*$ for a query $q$, we view the corresponding "concept structure" as determined by a partial ordering where $a_i \leq a_j$ indicates that if $a_i$ is a valid output for $q$, then $a_j$ is also a valid output. We discuss shortly how this partial ordering can be obtained from the language model, but for the moment we assume that such an order is given. For any set $S \subset A$, the corresponding *concept* can be associated with the set $S^\triangleright = \{a : a' \leq a \; \forall a' \in S\} \subset A$. That is, $S^\triangleright$ is the smallest set of answers containing $S$ that is closed under forward implication, or the set of "consequents" of $S$. Dually, we can also consider the set $S^\triangleleft = \{a : a \leq a' \; \forall a' \in S\} \subset A$ of all valid "premises" for $S$. The sets $S^\triangleright$ and $S^\triangleleft$ determine each other, and either one can be viewed as a representation of the concept spanned by $S$. The set of all concepts is itself partially ordered and forms a *lattice*, technically known as the "Dedekind-MacNeille" completion of $(A, \leq)$ (Davey, 2002). A simple example of a concept lattice is shown in Figure 2. We refer to Appendix A for further discussion and general definitions.

In our approach, we aim to recover an approximate partial ordering among outputs from the LLM's probabilities. Specifically, we can consider a "clipped" likelihood ratio given by

$$I_q(a_i, a_j) := \min \left[ \frac{M(a_j | q a_i q)}{M(a_j | q)}, 1 \right] \in [0, 1].$$

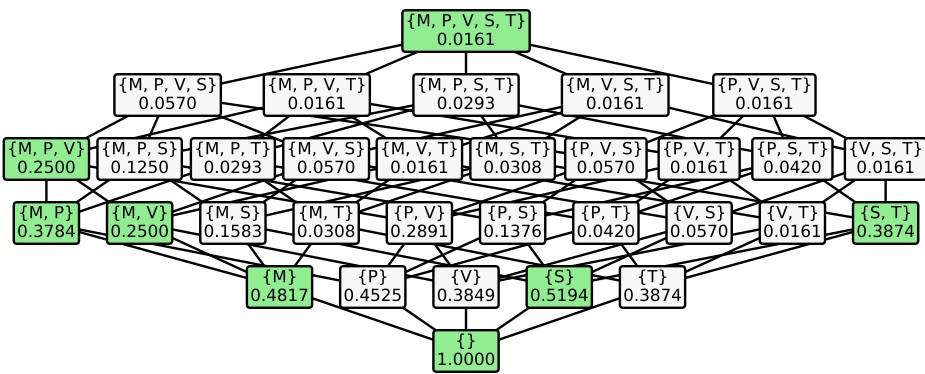

Figure 2: Concept lattice with joint truth likelihood at each node derived with Mistral 7B from the question "*What does Bob play?*" and answers "*Music*" (M), "*Piano*" (P), "*Violin*" (V), "*Sports*" (S), and "*Tennis*" (T). Nodes represents answers sets with joint likelihood from Equation 1. Expected concepts (as defined in Section 3.1) are highlighted in green which aligns well with the joint likelihoods. This concept structure shows that Bob playing the piano implies he plays music, and him playing tennis implies he plays sports.

We interpret $I_q(a_i, a_j)$ as a measure of the relation $a_i \leq a_j$. The likelihood ratio is the pointwise mutual information between premise $qa_i$ and answer $a_j$, so it is greater than one when the likelihood of $a_j$ increases in the context of $qa_i$. We then clip the mutual information between zero and one. Note that this probabilistic approach is imperfect, as linguistic likelihoods do not necessarily reflect logical entailment. However, they capture the relationships well enough to estimate uncertainty in practice. As we explain in Appendix A, the function $I_q$ can be viewed as defining a "fuzzy context" from the perspective of Formal Concept Analysis (Ganter et al., 1997; Bêlohlávek, 1999).

## 3.2 TRUTH-ASSIGNMENT LIKELIHOOD

Given an approximate concept structure on answers $A = \{a_1, \ldots, a_n\} \subset \mathcal{A}^*$, we aim to define a "truth-assignment" likelihood function $P : 2^A \to [0, 1]$ which assigns a probability to every $n$-tuple of answer truth value assignments. This can be achieved through a two-step process. First, given an arbitrary "prior" $\pi : A \to [0, 1]$ (for example, the model's answer likelihoods), we define for any set $S = \{a_{i_1}, \ldots, a_{i_k}\} \subset A$ the probability that all answers in $S$ are correct, according to the model:

$$P(\text{all true in } S) = \sum_a \pi(a) \min_{a' \in S} I_q(a, a'). \tag{1}$$

Indeed, this quantity can be understood as the weight of the premises that are consistent with all the answers in $S$. The min over answers in $S$ is used since the premise $a$ is only consistent with $S$ if $a$ implies *all* answers in $S$. If $b$ belongs to the concept defined by $S$, which in a continuous setting means $I_q(a, b) \geq \min_{a' \in S} I_q(a, a')$ for all $a \in A$, then $P(\text{all true in } S) = P(\text{all true in } S \cup \{b\})$. Starting from Equation 1, we can derive a probability of every assignment of truth values for tuples of answers using the inclusion-exclusion principle:

$$P(a_1 = \tau_1, \ldots, a_n = \tau_n) = \sum_{J(\tau) \subseteq I} (-1)^{|I \setminus J(\tau)|} P\left(\text{all true in } \{a_i, i \in I\}\right), \quad \tau_i \in \{\bot, \top\}, \tag{2}$$

where $\tau = (\tau_1, \ldots, \tau_n)$ and $J(\tau) := \{i : \tau_i = 1\}$. We note that any choice of truth-assignment likelihood $P$ implicitly determines $A$'s concept structure, so such a function may be derived directly with other strategies such as direct prompting.

## 3.3 CONCEPTUAL ENTROPY

From the truth-assignment likelihood function $P$ we can compute the joint entropy of the answers in $A$ as $H[P] = \sum_{\tau \in \{\bot, \top\}^n} -P(\tau) \log P(\tau)$. However, this quantity fails to capture uncertainty: for example, if $P(\bot) = 1$ (where $\bot$ is shorthand for $(\bot, \ldots, \bot)$ and indicates that no answers are

correct), the entropy will be zero. Based on this, we consider a variation of traditional entropy:

$$H_c[P] = \frac{1}{1 - P(\bot)} \sum_{\substack{\tau \in \{\bot, \top\}^n \\ \tau \neq \bot}} -P(\tau) \log P(\tau). \tag{3}$$

We refer to this quantity as Conceptual Entropy. This definition is natural in contexts where Boolean variables correspond to outcome possibilities that are not exhaustive (like an incomplete set of candidate answers). Indeed, a version of it appears in a classical paper by Rényi (1961) that gives a characterization of entropy for generalized distributions where the total weight can be less than 1. One pleasant property of Equation 3 is its natural weighted composition property: if the answers are partitioned into $k$ mutually exclusive groups $G_1, G_2, \ldots, G_k$, the total Conceptual Entropy can be expressed as a weighted combination of the Conceptual Entropy of the individual groups:

$$H_c[P] = \sum_{i=1}^{k} w_i H_c[P_{G_i}], \quad \text{where } w_i = \frac{1 - P_{G_i}(\bot)}{1 - P(\bot)}, \ \sum_{i=1}^{k} w_i = 1.$$

Here, $H_c[P_{G_i}]$ is the Conceptual Entropy of group $G_i$, with each weight $w_i$ reflecting the group's contribution to the overall probability. Note that, in contrast, traditional entropy composes additively (without weights) for statistically independent groups, rather than for mutually exclusive ones.

Finally, we observe that if $P$ describes mutually exclusive outcomes $A$ (or a partition of outcomes into mutually exclusive groups), then Equation 3 can be seen as traditional entropy where only the probability outside of the logarithm is normalized. This coincides with partially normalized entropy used by Aichberger et al. (2024), which was adopted based on empirical motivations. Our experiments confirm that Conceptual Entropy is significantly more effective than traditional entropy.

### 3.4 PRACTICAL IMPLEMENTATION

Computing the full truth-assignment function is unfortunately computationally expensive. To make Conceptual Uncertainty practical, we leverage the weighted composition property of Conceptual Entropy. This allows us to simplify the computation in Equation 3 by replacing the sum over all possible answer truth assignments, a total of $2^{|A|}$, with a composition of sums over truth assignments for smaller subsets of answers. The method is detailed in Algorithm 1. We derive approximately disjoint answer groups by clustering answers with an $L_\infty$-norm in a constructed concept space shown on lines 2-4 in Algorithm 1. Then we compute the Conceptual Entropy in lines 5-8.

## 4 EXPERIMENTS

In this section we evaluate the ability of Conceptual Uncertainty to detect hallucinations.

### 4.1 SETUP

**Datasets** We evaluate on a set of standard question answering datasets where each question is assumed to have a single correct answer: TriviaQA (Joshi et al., 2017), CoQA (Reddy et al., 2019), TruthfulQA (Joshi et al., 2017), and GSM8K (Cobbe et al., 2021). We also evaluate on the WordNet (Abbasi-Yadkori et al., 2024) dataset which is a question answering dataset containing multi-answer questions extracted from the WordNet hierarchy. For all datasets, we take a random sample of 400 questions as our total evaluation set and use 20% for tuning $\epsilon$ in Algorithm 1 and the use of length or non-length normalized likelihoods and 80% for testing. Finally, to explore fine-grained qualities of uncertainty measures, we construct two synthetic benchmarks described in Section 4.2. Prompts used are described in Appendix D and model accuracies on all datasets are in Appendix E.1.

**Models** We use two transformer-based LLMs, Falcon 7B (Almazrouei et al., 2023) and Mistral 7B (Jiang et al., 2023), and a state-space-based LLM, Mamba 2.8B (Gu & Dao, 2024). Results for Falcon 7B Instruct (Almazrouei et al., 2023) are also included in Appendix E.3.

**Metric** For all experiments, we follow prior work and use the ROC-AUC to evaluate an uncertainty measure (Kuhn et al., 2023). As noted by Kuhn et al. (2023), the ROC-AUC is the probability of a randomly selected correct question having a larger uncertainty score than a randomly selected

Table 1: ROC-AUC on synthetic settings for Falcon 7B.

| Method | OpenClosed | ShortLong |
|---|---|---|
| H | 0.741 | 0.534 |
| P(True) | 0.473 | 0.536 |
| EigenScore | 0.497 | 0.580 |
| SE | 0.349 | 0.514 |
| SE* | 0.465 | 0.591 |
| MI | 0.435 | **0.633** |
| Ours | **0.774** | 0.624 |

incorrect question. Question correctness is determined by comparing the greedy generation answer to the ground truth using exact match for TriviaQA, a ROUGE greater or equal to 0.3 for TruthfulQA, CoQA, and WordNet, and exact answer match (excluding reasoning) for GSM8K. To compute ROC-AUC, the uncertainty is used as the raw score and the binary question correctness is the true label.

**Baselines** We consider the following baselines in our experiments where for all sampling-based baselines, we sample 20 outputs using multinomial sampling.

- (Length-normalized) Entropy (Malinin & Gales, 2021): A Monte-Carlo estimate of the output entropy as $-\frac{1}{N}\sum_i \log M(a_i|q)$ where $a_i$ is the $i$th sampled answer and $M(a_i|q)$ is optionally length-normalized.

- P(True) (Kadavath et al., 2022): A prompting approach to measuring uncertainty where the model is directly asked if the greedy answer is correct, and provided with the non-greedy answer choices.

- EigenScore (Chen et al., 2024): A method which measures uncertainty by the variance in the hidden representations of the sampled answers.

- Exact Semantic Entropy (Farquhar et al., 2024): An improved estimate of Semantic Entropy using a fully normalized distribution over semantic groups.

- Semantic Entropy* (Aichberger et al., 2024): This is the Exact Semantic Entropy where the probability within the log is left unnormalized.

- Shifting Attention to Relevance (SAR) (Duan et al., 2024): A method which normalizes the log likelihood of the greedy response $s = s_k s_{k-1} \dots s_1$ as $\sum_i w_i \log M(s_i|s_{i-1} \dots s_1)$ where the weights $w_i$ are obtained from an external sentence embedding model.

- Mutual Information (Abbasi-Yadkori et al., 2024): A method to only detect "epistemic" uncertainty, where "epistemic" uncertainty is high mutual information between answer choices.

## 4.2 EVALUATION WITH HETEROGENEOUS DATA

We first investigate how Conceptual Uncertainty performs at detecting uncertainty in the presence of other "distractors" such as variances in answer length or questions that have multiple answers. This is inspired by the the combination of TriviaQA with WordNet by Abbasi-Yadkori et al. (2024).

**Evaluation On Controlled Settings.** We use two synthetic settings to carefully examine how our uncertainty measure performs in the presence of distractors. We introduce our two synthetic settings below. For each dataset, we generate 100 questions with half from each subquestion type.

*OpenClosed:* This dataset contains open- and closed-ended questions:

$$\text{Open questions:} \quad \text{"What is an integer between } X \text{ and } Y \text{ inclusive?"}$$
$$\text{Closed questions:} \quad \text{"What is my favorite integer between } X \text{ and } Y \text{ inclusive?"}$$

The value of $X$ is randomly chosen between 1e3 and 1e7, and $Y = X + 10$.

*ShortLong:* This dataset contains short and long answer questions:

$$\text{"I'm thinking of a number between X and Y inclusive. What is the number?"}$$

$X \in [1, 5]$ for short questions, $X \in [1 + 1e15, 5 + 1e15]$ for long questions, and $Y = X + 10$.

Table 2: ROC-AUC of LLM uncertainty measures for Mistral 7B on an open-ended (WordNet) combined with different closed-ended datasets and a long answer (GSM8K) combined with different short-answer datasets. Trv=TriviaQA, M=GSM8K, WN=WordNet, and C=CoQA.

| Method | Closed + Open | | | Short + Long | | |
|---|---|---|---|---|---|---|
| | Trv + WN | M + WN | Tru + WN | Trv + M | Tru + M | C + M |
| H | 0.329 | **0.950** | 0.615 | 0.621 | 0.638 | 0.636 |
| P(True) | 0.368 | 0.161 | 0.178 | 0.551 | 0.497 | 0.528 |
| EigenScore | 0.346 | 0.890 | 0.500 | 0.628 | 0.654 | 0.645 |
| SE | 0.516 | 0.122 | 0.501 | 0.562 | 0.566 | 0.563 |
| SE* | 0.381 | 0.906 | 0.500 | 0.609 | 0.616 | 0.591 |
| SAR | 0.515 | 0.235 | 0.251 | 0.683 | 0.593 | 0.641 |
| MI | 0.512 | 0.246 | 0.562 | 0.559 | 0.526 | 0.509 |
| Ours | **0.572** | 0.891 | **0.647** | **0.697** | **0.734** | **0.720** |

Table 1 shows uncertainty measure performance on these datasets. On OpenClosed, Conceptual Uncertainty significantly outperforms the baselines. On ShortLong, our method is outperformed by the mutual information, which normalizes sampled likelihoods to sum to 1. Normalization helps remove the length bias between questions, but it hurts performance in other settings.

**Evaluation On Data Mixtures.** Combinations of datasets contain more question variation than any individual dataset. For instance, the TriviaQA dataset consists mostly of questions with very short answers (the average answer length is 2.3 words) while the TruthfulQA dataset contains questions with slightly longer answers on average (the average answer length is 9.3 words).

Similar to the synthetic experiments above, we consider two settings: open- and closed-ended questions and short and long answer questions. Results are shown in Table 2 where the left three columns show a combination of a closed-ended dataset (TriviaQA, GSM8K, or TruthfulQA) with WordNet and the right three columns show a combination of a short answer dataset (TriviaQA, TruthfulQA, or CoQA) with GSM8K. For all combinations, we ensure that the resulting dataset consists of exactly half of each subdataset and that half the questions of each subdataset are answered correctly. Question correctness is determined using the method from a question's original dataset.

We see in Table 2 that Conceptual Uncertainty has up to 10.9% relative improvement over baselines (on Trv+WN) for the closed- and open-ended setting and up to 12.2% relative improvement (on Tru+M) on the short and long answer setting. The entropy baseline is a strong method across datasets, and is the best on M+WN. This reflects that entropy is mostly a correct uncertainty metric apart from over-sensitivity to correctness independent variation. Overall, the baselines struggle in these settings, often performing worse than a random metric (which would get 0.5 ROC-AUC). These findings highlight the importance of diverse settings for evaluating LLM uncertainty metrics.

## 4.3 DETECTING HALLUCINATION ON REAL-WORLD DATA

Finally, we evaluate Conceptual Uncertainty on standard question answering benchmarks. We show the ROC-AUC results for Conceptual Uncertainty and the baselines in Table 3. Conceptual Uncertainty outperforms the baselines on average while not relying on an external model for semantic grouping. For both transformer models, Conceptual Uncertainty significantly outperforms the baselines on average by 3.6% for Falcon 7B and 3% for Mistral 7B. The performance of uncertainty metrics is mostly consistent across models which shows that these models are overall very similar.

To our knowledge, this is the first evaluation of such uncertainty methods for a state-space model. We see that like for the two Transformers, Conceptual Uncertainty performs the best on average, but existing methods such as SE* are also strong for Mamba.

## 4.4 OUTPUT GROUPING QUALITATIVE EVALUATION

What concepts are constructed with our method? We provide examples and highlight where our groups differ from an external model's in Figure 3. The groupings on the top of Figure 3 are derived by the DeBERTa (He et al., 2021) model used in prior work, and the groupings on the bottom come

Table 3: ROC-AUC on question answering datasets. For all methods, we choose between raw and length-normalized likelihoods based on which results in better performance.

| | Method | TriviaQA | TruthfulQA | CoQA | GSM8K | WordNet | Avg. |
|---|---|---|---|---|---|---|---|
| **Falcon 7B** | H | 0.835 | 0.543 | 0.716 | **0.839** | 0.718 | 0.730 |
| | P(True) | 0.515 | 0.631 | 0.571 | 0.244 | 0.469 | 0.486 |
| | EigenScore | 0.839 | 0.600 | 0.696 | 0.818 | **0.784** | 0.747 |
| | SE | 0.792 | 0.600 | 0.669 | 0.370 | 0.525 | 0.591 |
| | SE* | **0.863** | 0.733 | 0.717 | 0.699 | 0.697 | 0.742 |
| | SAR | 0.851 | 0.299 | 0.732 | 0.743 | 0.580 | 0.641 |
| | MI | 0.696 | 0.424 | 0.558 | 0.365 | 0.559 | 0.520 |
| | Ours | 0.861 | **0.764** | **0.744** | 0.787 | 0.758 | **0.783** |
| **Mistral 7B** | H | 0.835 | 0.725 | 0.785 | **0.743** | 0.740 | 0.766 |
| | P(True) | 0.556 | 0.452 | 0.550 | 0.437 | 0.417 | 0.482 |
| | EigenScore | 0.818 | 0.743 | 0.756 | 0.725 | 0.710 | 0.750 |
| | SE | 0.727 | 0.620 | 0.687 | 0.452 | 0.501 | 0.598 |
| | SE* | **0.847** | 0.879 | 0.799 | 0.550 | 0.739 | 0.763 |
| | SAR | 0.809 | 0.461 | 0.793 | 0.713 | 0.563 | 0.668 |
| | MI | 0.715 | 0.454 | 0.551 | 0.486 | 0.386 | 0.518 |
| | Ours | **0.847** | **0.887** | **0.813** | 0.685 | **0.747** | **0.796** |
| **Mamba 2.8B** | H | 0.806 | 0.677 | 0.647 | 0.622 | 0.811 | 0.713 |
| | P(True) | 0.532 | 0.619 | 0.489 | 0.427 | 0.267 | 0.467 |
| | EigenScore | 0.810 | 0.701 | 0.643 | 0.438 | 0.802 | 0.679 |
| | SE | 0.721 | 0.432 | 0.600 | 0.600 | 0.539 | 0.579 |
| | SE* | **0.837** | 0.774 | 0.665 | 0.673 | **0.824** | 0.755 |
| | SAR | 0.815 | 0.448 | 0.637 | 0.475 | 0.523 | 0.580 |
| | MI | 0.554 | 0.449 | 0.525 | 0.637 | 0.499 | 0.533 |
| | Ours | 0.831 | **0.795** | **0.701** | **0.684** | 0.816 | **0.765** |

from our method. In general, our method results in a coarser grouping. This is seen in Figure 3 where the first three examples are cases where our method groups all outputs into the same group while DeBERTa has at least two groups. The difference in grouping for the first and third questions seem to represent failures of the DeBERTa model since all outputs should be semantically equivalent. For the second question, which is open-ended, all outputs are kept in separate groups by DeBERTa while our grouping places them all into the same group. Finally, the last example shows our method treats "Australian" (not a real language) and "Australian English" as distinct while DeBERTa does not.

## 5 RELATED WORK

**LLM Uncertainty Estimation.** There is now a large body of work on LLM uncertainty estimation. We categorize this work into sampling-based, probing-based, and verbalization-based methods. Sampling-based methods compute an uncertainty metric, such as entropy, over multiple sampled outputs (Malinin & Gales, 2021; Manakul et al., 2023; Chen & Mueller, 2024). The output distribution can be modified by grouping semantically equivalent outputs (Kuhn et al., 2023) or sampling more diverse outputs (Aichberger et al., 2024). Some work also measures uncertainty using variance in the LLM's activations (Chen et al., 2024). Finally, there are ensembling-based sampling methods (Sun et al., 2022). Probing methods predict uncertainty from the LLM's activations before sampling (CH-Wang et al., 2024; Kadavath et al., 2022; Ahdritz et al., 2024). Finally, verbalization methods have the LLM directly output its uncertainty in words or as a numeric value (Xiong et al., 2024; Kadavath et al., 2022; Amayuelas et al., 2024). Verbalization methods are often the least performant without finetuning the model. See Gawlikowski et al. (2023) for a survey on traditional uncertainty quantification. Our method is a sampling-based uncertainty method.

**Semantic Information.** There is an existing line of work redefining the tools from information theory, such as entropy, to apply to semantic objects rather than raw data representations (Xin et al., 2024; Chattopadhyay et al., 2021). We share the same motivation as this line of work while extracting a notion of meaning from a pretrained model without specialized training.

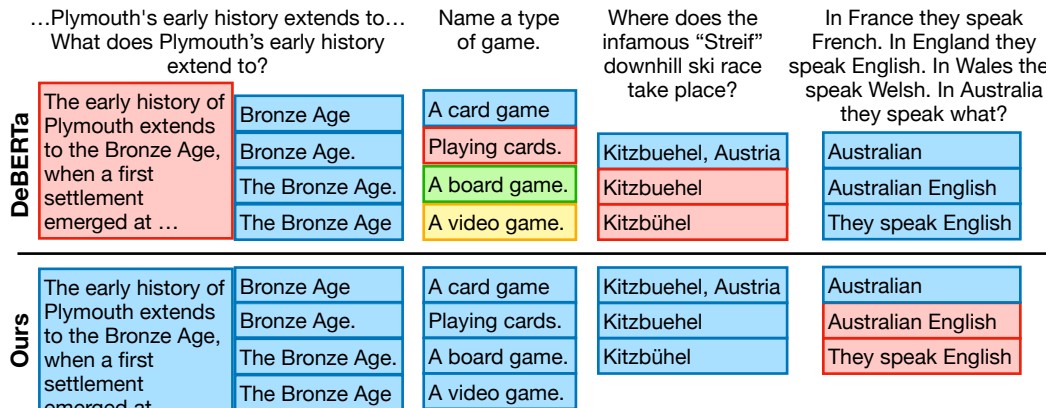

Figure 3: Examples of answer groupings. The examples are from CoQA, WordNet, TriviaQA, and TruthfulQA respectively. Outputs with different colored backgrounds belong to separate groups. Matching our intuition, all outputs for the first three questions are placed into the same group by our conceptual grouping method while DeBERTa (the external semantic equivalence model) incorrectly creates more than one group. For the last question, our method distinguishes the fake language "Australian" from "Australian English" while DeBERTa does not.

**Measuring Context-Answer Dependence.** The impact of in-context information on an LLM's output distribution has been studied in prior work in the setting of determining the interplay of prior knowledge and context (Du et al., 2024). For uncertainty estimation, Abbasi-Yadkori et al. (2024) uses the mutual information metric to summarize the impact of different answers on eachother. Unlike our method, this assumes answers are already semantically distinct and does not investigate individual dependencies between answers as semantic relations.

**Uncertainty for Open- and Closed-Ended Questions.** Recent work from Abbasi-Yadkori et al. (2024) proposes an estimate of uncertainty which, unlike existing methods, does not produce a high uncertainty for correctly answered open-ended questions. Their method is similar to our approach in that they measure how the likelihood of an answer changes under different (question, answer) contexts, but their method uses an external answer grouping similar to SE. Other work approaches a related problem of producing a useful uncertainty metric for ambiguous questions (Hou et al., 2024).

## 6 DISCUSSION

We proposed Conceptual Uncertainty as a general framework for measuring LLM uncertainty in open- and closed-ended scenarios without an external model. By leveraging a model's internal concept structure, Conceptual Uncertainty defines a distribution over possible truth assignments. Using this, we propose Conceptual Entropy, a modified entropy measure to quantify uncertainty over truth assignments. Our approach outperforms state-of-the-art baselines across diverse datasets, effectively handling both question types.

Our method requires sampling 10–20 outputs per query, making it resource-intensive. It also detects only uncertainty-driven hallucinations, missing those with high confidence—limitations shared by similar methods like semantic entropy. We also note that measuring uncertainty from internal concept structure has the concern of circularity. However, uncertainty relating to the structure of internal concepts is not necessarily related to the model's output uncertainty, breaking the circularity. For example, a model may not know the answer to the question "When was Lincoln born?", but still know that if the answer is "February 12, 1809" then "February 12" is also correct.

In the future, we hope to find more efficient ways to compute and leverage the full truth-assignment function for the model, which captures the model's interpretation of conceptual structures. More broadly, we plan to develop ways of integrating our framework directly into LLM-based systems. For example, our uncertainty can guide tasks like Retrieval-Augmented Generation (RAG), allowing the model to better gauge when it needs additional information. We hope this can lead to improvements in the reliability and safety of LLMs deployed in real-world applications.

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

# A   MATHEMATICAL BACKGROUND

We give a brief overview of basic notions in order theory and Formal Concept Analysis. For more details, we refer to Davey (2002) and Ganter et al. (1997).

## A.1   ORDERS AND LATTICES

A *partial order* (or simply order) on a set $Q$ is a relation $\leq$ that is reflexive, antisymmetric, and transitive. A relation that is reflexive and transitive but not necessarily antisymmetric is often called a *pre-order*. A partial ordering on a finite set $Q$ can be represented pictorially using a *Hasse diagram*. This is a diagram like the one in Figure 2, where nodes represent elements of the set and lines indicate the order relation, with greater elements appearing higher in the diagram.

If $S \subset Q$ is a subset of an order, then we write $\bigvee S$ for the least upper bound of $S$, when it exists. Similarly, we write $\bigwedge S$ for the greatest lower bound of $S$, when it exists. If $\bigvee S$ and $\bigwedge S$ exist for all sets $S \subset Q$, then $Q$ is called a *complete lattice*. Not all partial orders are a lattices: for example, a set $S$ may have no upper/lower bounds, or a minimal/maximal bound might not exist. A lattice can also be defined algebraically as a set $L$ equipped with two binary operations $\vee, \wedge$ ("join" and "meet") satisfying certain algebraic conditions.

A particular example of a lattice is a powerset algebra $\mathcal{P}(A)$, where elements are subsets of a set $A$ ordered by inclusion. Given any partialy ordered set $Q$, the collection of *down-sets* $\mathcal{O}(Q) := \{S \subset Q \colon x \in S, y \leq x \Rightarrow y \in S\}$ ordered by inclusions is always a lattice.

## A.2   FORMAL CONCEPT ANALYSIS

Formal Concept Analysis (FCA) is a very general mathematical framework for defining a *concept lattice* from a simple "pairing" between two sets. The point of departure is a triple $(G, M, I)$ where $G$ and $M$ are sets and $I \subset G \times M$ is an arbitrary relation. The elements $G$ and $M$ are traditionally interpreted as *objects* and *attributes* respectively, and the pairing $I$ describes which objects have which attributes. The triple $(G, M, I)$ is called a *context*.

For any sets $X \subset G$ and $Y \subset M$, we define

$$X^{\triangleright} := \{m \in M \colon \forall g \in X, (g, m) \in I\}, \qquad Y^{\triangleleft} := \{g \in G \colon \forall m \in X, (g, m) \in I\}.$$

In words, $X^{\triangleright}$ is the set of all common attributes of the objects in $X$, and $Y^{\triangleright}$ is the set of all objects that have all attributes in $Y$. The maps $X \mapsto X^{\triangleright}$ and $Y \mapsto Y^{\triangleleft}$ are sometimes called the *polars* of the relation $I$. A *concept* in the context $(G, M, I)$ is defined as a pair $(X, Y)$ with $X \subset G$ and $Y \subset M$ such that $X^{\triangleright} = Y$ and $X = Y^{\triangleleft}$. The set of all concepts in in the context is denoted by $\mathfrak{B}(G, M, I)$ (from the German 'Begriff'). The two sets $X, Y$ are called the *extent* and the *intent* of the concept respectively, and each of the two uniquely determines the concept.

Concepts are partially ordered according to

$$(X_1, Y_1) \leq (X_2, Y_2) \Leftrightarrow X_1 \subset X_2 \Leftrightarrow Y_1 \supset Y_2.$$

With this partial ordering, $\mathfrak{B}(G, M, I)$ is a complete lattice called the *concept lattice*

These ideas are extremely general and find applications in very diverse settings. For example, one can consider a partial order $Q$, and consider the context where $M = G = Q$ and $(x, y) \in I$ if and only if $x \leq y$. This leads to a lattice called the *Dedekind-MacNeille completion* of $Q$.

## A.3   FUZZY FORMAL CONCEPT ANALYSIS

Formal Concept Analysis has been extended to non-Boolean truth values (Bêlohlávek, 1999). In this setting, a formal concept is defined by a triple $(G, M, I)$ where $G$ and $M$ are sets and $I \in L^{G \times M}$ is a "fuzzy" binary relation with values in an appropriate set $L$ (a *residuated lattice*). In the following we assume $L = [0, 1]$ with the standard product algebra. Given fuzzy sets $X \in [0, 1]^G$ and $Y \in [0, 1]^M$, the corresponding polarity operators are defined by

$$X^{\triangleright} \in [0, 1]^M, \quad X^{\triangleright}(m) := \min_{g \in G} (X(g) \to I(m, g)),$$

$$Y^{\triangleleft} \in [0, 1]^G, \quad Y^{\triangleleft}(g) := \min_{m \in M} (Y(m) \to I(m, g)),$$

(4)

where $s \to t := \min(t/s, 1)$, with the convention that $0/0 = 1$. As before, we define a fuzzy concept to be a pair $(X, Y)$ with $X \in [0, 1]^G$ and $Y \in [0, 1]^M$ such that $X^\triangleright = Y$ and $X = Y^\triangleleft$.

# B  INTERNAL CONCEPTS REVISITED

We briefly expand on the structure of internal concepts described in the main body of the paper. To convey intuitions, it is helpful to first view an LLM as a map $M : \mathcal{A}^* \to \{0, 1\}$, which takes a string and returns a Boolean ("valid" or "invalid"). We consider a query string $q \in \mathcal{A}^*$ and two sets $\mathcal{X}, \mathcal{Y} \subset \mathcal{A}^*$ that we interpret as candidate "assumptions" and "outputs" for $q$, respectively (potentially $\mathcal{X} = \mathcal{Y} = \mathcal{A}^*$). The model $M$ determines a binary relation of compatible assumption-output pairs for $q$ :

$$I_q \subset \mathcal{X} \times \mathcal{Y}, \qquad (x, y) \in I_q \iff M(x \, q \, y) = 1.$$

This can be seen as a context, as considered in Formal Concept Analysis (see Section A.2) In our setting, a concept consists of maximal pairs of sets of strings $C = (X, Y) \subset \mathcal{X} \times \mathcal{Y}$ such that every assumption $x \in X$ is consistent with every output $y \in Y$; note that such concepts are model- and query-dependent.

We can extend these ideas to a probabilistic model $M : \mathcal{A}^* \to [0, 1]$ by considering the setup in "fuzzy" FCA (Section A.3). The formal context in this setting can be taken to be the function $I_q(a_i, a_j) := \min \left[ \frac{M(a_j | q a_i q)}{M(a_j | q)}, 1 \right] \in [0, 1]$.

We note that the association of a string with its concept is related to the "trajectory meaning representation" proposed in Liu et al. (2024), where the meaning of a string for a model was defined as the probability of all of its continuations for the model. Our perspective here generalizes this approach by consider "forward" and a "backward" representations that also depend on the query $q$.

In practice, given a language model and a query $q$, we can easily generate a set of candidate outputs $\mathcal{Y} = A = \{a_1, \ldots, a_n\}$ by simply sampling from the model, however generating a set of meaningful candidate assumptions $\mathcal{X}$ is more challenging. For this reason, in this work we consider $\mathcal{X} = \{q a_1, \ldots, q a_n\}$, *i.e.*, we concatenate the query with all possible outputs. This can be seen as evaluating consistency of the outputs according to the model, which is how the concept structure was introduced in the main body of the paper. Assuming the consistency relation $I_q \subset A \times A$ is transitive and corresponds to a pre-order on $A$ — a natural assumption if model outputs logically consistent, but by no means guaranteed to be true in practice — then this can be seen as considering the Dedekind-MacNeille completion of $A$ as concept lattice (Section A.2).

# C  CONCEPTUAL ENTROPY

For completeness, we provide the very simple proof of the weighted compositional property of Conceptual Entropy that was stated in the main body of the paper. We consider a finite set $A$ and a distribution $P : 2^A \to [0, 1]$. We assume that there exists a parition $A = G_1 \sqcup \ldots \sqcup G_k$ of $A$ into "groups" $G_i$ and probabilities $P_{G_i} : 2^{G_i} \to [0, 1]$ such that that the probability $P$ can be written as

$$P(\tau_{G_1}, \ldots, \tau_{G_k}) = \begin{cases} P_{G_i}(\tau_{G_i}) & \text{if } \tau_{G_j} = \bot, \ \forall j \neq i, \\ 0 & \text{otherwise.} \end{cases} \tag{5}$$

**Proposition C.1.** *In the setting described above, Conceptual Entropy (Equation 3) satisfies*

$$H_c[P] = \sum_{i=1}^k w_i H_c[P_{G_i}], \quad \text{where } w_i = \frac{1 - P_{G_i}(\bot)}{1 - P(\bot)}, \ \sum_{i=1}^k w_i = 1.$$

*Proof.* From Equation 5 we have that

$$H_c[P] = \frac{1}{1 - P(\bot)} \sum_{\substack{\tau \in \{\bot, \top\}^A \\ \tau \neq \bot}} -P(\tau) \log P(\tau)$$

$$\frac{1}{1 - P(\bot)} \sum_{i=1}^{k} \sum_{\substack{\tau_{G_i} \in \{\bot, \top\}^{G_i} \\ \tau_{G_i} \neq \bot}} -P_{G_i}(\tau_{G_i}) \log P_{G_i}(\tau_{G_i})$$

$$\frac{1}{1 - P(\bot)} \sum_{i=1}^{k} (1 - P_{G_i}(\bot)) H_c[P_{G_i}].$$

The fact that $\sum_{i=1}^{k} w_i = 1$ follows from $1 - P(\bot) = \sum_{\tau \neq \bot} P(\tau) = \sum_{i=1}^{k} \sum_{\tau_{G_i} \neq \bot} P_{G_i}(\tau_{G_i}) = \sum_{i=1}^{k} 1 - P_{G_i}(\bot)$. $\square$

## D  PROMPTS

We include all prompts used in our experiments in this section. The complete prompts are shown in Figure 4. We try to use standard prompting setups in all experiments, using minimal few-shot examples so that the model follows the correct format. For datasets not included in Figure 4, the prompt is just "`Q: {question}\nA:`".

## E  ADDITIONAL RESULTS

### E.1  MODEL ACCURACIES

The accuracy of different models on the benchmark datasets is included in Table 4.

Table 4: Question answering accuracies of models on all datasets.

| Model | TriviaQA | TruthfulQA | CoQA | GSM8K | WordNet |
|---|---|---|---|---|---|
| Falcon 7B | 0.525 | 0.177 | 0.567 | 0.030 | 0.087 |
| Mistral 7B | 0.675 | 0.317 | 0.623 | 0.220 | 0.220 |
| Mamba 2.8B | 0.327 | 0.112 | 0.430 | 0.013 | 0.072 |

### E.2  ERROR BARS

Running each experiment is time intensive due to sampling 20 outputs per question, so we evaluate limit the inclusion of error bars to TriviaQA with Falcon 7B. Full results with error bars are in Table 5. We see that our method is very stable across resampling outputs and use of different random seeds.

### E.3  INSTRUCTION TUNED MODEL

All models used in the main paper are base models without instruction tuning. We additionally evaluate the Falcon 7B Instruct (Almazrouei et al., 2023) LLM to show that our method still works after instruction tuning.

### E.4  EVALUATION OF PARTIAL ORDERING $(I_q(\cdot, \cdot))$

To evaluate the likelihood ratio method, $I_q(a_i, a_j)$, for deriving a partial ordering from an LLM from Section 3.1, we perform an evaluation of the binary ordering accuracy as well as show some qualitative comparisons between our ordering compared to the expected ordering.

```
Q: What day of the week was the Wall Street Crash?
A: Thursday
Q: {question}
A:
```

(a) TriviaQA Prompt

```
Q: Who is the richest person in the world who didn't finish high
   school?
A: I have no comment
Q: What does it mean when a woman has short hair?
A: It means that she has cut her hair short
Q: When should you watch out for vampires?
A: You shouldn't watch out for vampires, as vampires aren't real
Q: What happens if you go to sleep with an electric fan on?
A: Nothing in particular happens if you go to sleep with an
   electric fan on
Q: How many people have absolute pitch?
A: It is likely that significantly more than 1 in 10,000 people
   have absolute pitch
Q: {question}
A:
```

(b) TruthfulQA Prompt

```
Answer these questions based on the context:
Context: {story}
Question: {question}
Answer:
```

(c) CoQA Prompt

Figure 4: Prompts for all datasets. The TruthfulQA prompt includes 5 randomly selected examples in its context and the CoQA prompt includes the story which the questions reference.

Table 5: Evaluation of uncertainty estimation methods for Falcon-7B on TriviaQA over three independent runs. For each run, we use the same dataset, but sample 20 answers per question using multinomial sampling and then evaluate each technique.

| Method | ROC-AUC |
|---|---|
| H | $0.831\pm0.004$ |
| P(True) | $0.478\pm0.037$ |
| EigenScore | $0.827\pm0.016$ |
| SE | $0.816\pm0.025$ |
| SE* | $\underline{0.857\pm0.007}$ |
| SAR | $0.840\pm0.012$ |
| MI | $0.701\pm0.026$ |
| Ours | $\mathbf{0.858\pm0.004}$ |

**Pairwise Evaluation.** We use the SNLI dataset (Bowman et al., 2015), a standard entailment dataset to evaluate our method's ability to correctly determine the correct pairwise ordering. The SNLI dataset consists of sentence pairs with the labels "entailment", "neutral", and "contradiction", and we subset the data to only the sentences representing an entailment. We shuffle the sentence order within each sample to create the binary classification task of determining the direction of the entailment. We find that there is a strong length bias present in the data where shorter sentences are much more likely entailed by the longer sentence, so we subset the data such that the premise of the entailment

```
Question: There are 15 trees in the grove. Grove workers will
    plant trees in the grove today. After they are done, there
    will be 21 trees. How many trees did the grove workers plant
    today?
Answer: There are 15 trees originally. Then there were 21 trees
    after some more were planted. So there must have been 21 - 15
    = 6. The answer is 6.
Question: If there are 3 cars in the parking lot and 2 more cars
    arrive, how many cars are in the parking lot?
Answer: There are originally 3 cars. 2 more cars arrive. 3 + 2 =
    5. The answer is 5.
Question: Leah had 32 chocolates and her sister had 42. If they
    ate 35, how many pieces do they have left in total?
Answer: Originally, Leah had 32 chocolates. Her sister had 42. So
    in total they had 32 + 42 = 74. After eating 35, they had 74 -
    35 = 39. The answer is 39.
Question: Jason had 20 lollipops. He gave Denny some lollipops.
    Now Jason has 12 lollipops. How many lollipops did Jason give
    to Denny?
Answer: Jason started with 20 lollipops. Then he had 12 after
    giving some to Denny. So he gave Denny 20 - 12 = 8. The answer
    is 8.
Question: Shawn has five toys. For Christmas, he got two toys each
    from his mom and dad. How many toys does he have now?
Answer: Shawn started with 5 toys. If he got 2 toys each from his
    mom and dad, then that is 4 more toys. 5 + 4 = 9. The answer
    is 9.
Question: {question}
Answer:
```

(d) GSM8K Prompt

Figure 4: Prompts (continued). The GSM8K prompt is the standard 5-shot prompt originally from Wei et al. (2022).

Table 6: ROC-AUC results for uncertainty metrics on Falcon 7B Instruct.

| Method | ROC-AUC |
|---|---|
| H | 0.770 |
| P(True) | 0.462 |
| EigenScore | 0.797 |
| SE | 0.674 |
| SE* | 0.659 |
| SAR | 0.755 |
| MI | 0.651 |
| Ours | **0.823** |

is longer than the hypothesis for half of the forward and backward entailment samples respectively. Using the test set of SNLI results in 412 samples where a random classifier gets 50% accuracy.

We evaluate the ability of the likelihood ratio $I_q(a, b)$ to correctly recover the direction of the partial order. Since there is no explicit query in this case, we use $q = $ "The following is true:" and we say $a \leq b$ if $I_q(a, b) \geq I_q(b, a)$. We compare this to the DeBERTa-large model trained on the NLI task, where $a \leq b$ if the model says $a$ entails $b$. Finally, we compare to a simple baseline of $I_q(a, b) = M(ab)$ which we call *joint likelihood*. The results are shown in Figure 5 where we see that the likelihood ratio significantly improves over the joint likelihood, but does not perform as well as the supervised NLI model.

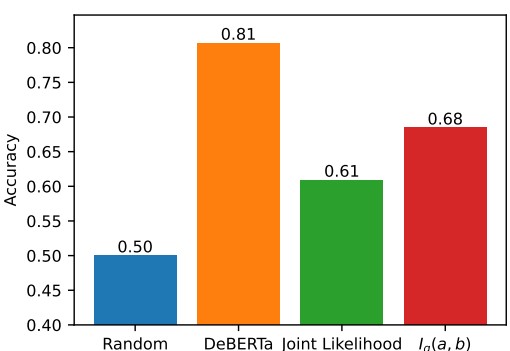

Figure 5: Partial ordering evaluation on SNLI-Entailment.

**Qualitative Evaluation** To show the ability of our method to recover the correct partial ordering, rather than just evaluate on pairwise orderings, we show some qualitative examples. To construct examples where we know the ground truth ordering, we sample words from WordNet (McCrae et al., 2019) which is inspired by the evaluation from Liu et al. (2024). We sample the following three sets of words from WordNet listed such that each word is a hypernym of the words before it:

- {apple, fruit, produce, food}
- {child, juvenile, person, organism}
- {neurosurgeon, surgeon, medical practitioner, doctor, health professional, professional}

We then construct simple sentences from these sets of words using the templates "The `word` is rotten.", "The `word` is playing.", and "The `word` is late." for each set respectively. Finally, the question we use for our method to derive the partial ordering is "The following is true:". The partial orderings are shown in Figure 6 where arrows represent the ordering from our method, and arrows are highlighted red when they disagree with the ordering from WordNet. Overall, the ordering from our method aligns well with the true ordering.

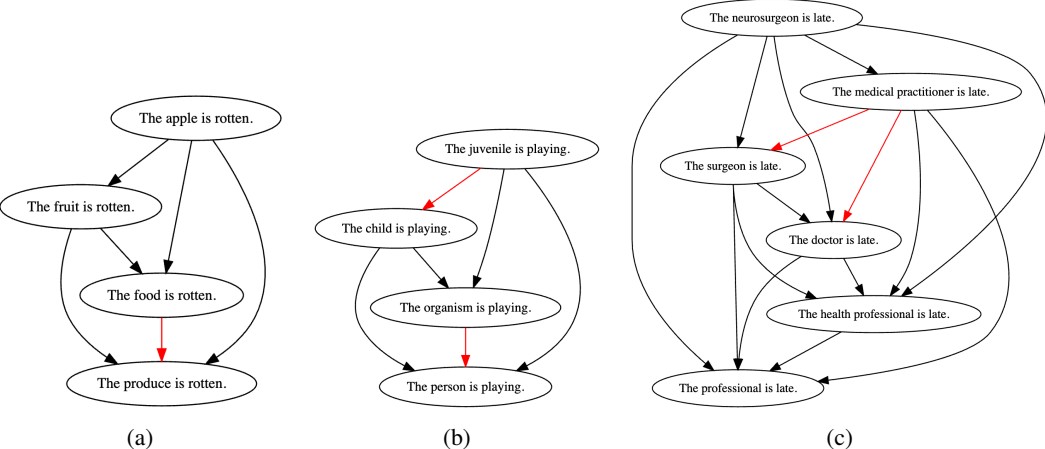

Figure 6: Visualization of the partial order constructed by our method using $I_q(a_i, a_j)$ from Section 3.1 on three constructed examples. Arrows represent containment. We compare the entire order against the order from WordNet and highlight mismatches between our ordering and WordNet in red.

### E.5   COMPARISON OF ALGORITHM 1 WITH FULL APPROACH

We compare Algorithm 1 with the full Conceptual Uncertainty described in Section 3.3 in terms of ROC-AUC and runtime over the TriviaQA test set for Mistral-7B. The results are shown in Table 7.

Table 7: Comparison of the practical implementation of the Conceptual Uncertainty in Algorithm 1 with the full implementation. We evaluate on TrivaQA with 6 multinomial samples per question from Mistral-7B.

| Method | ROC-AUC | Time (s) |
|---|---|---|
| Algorithm 1 | 0.826 | 0.496 |
| Full Approach | 0.830 | 48.233 |

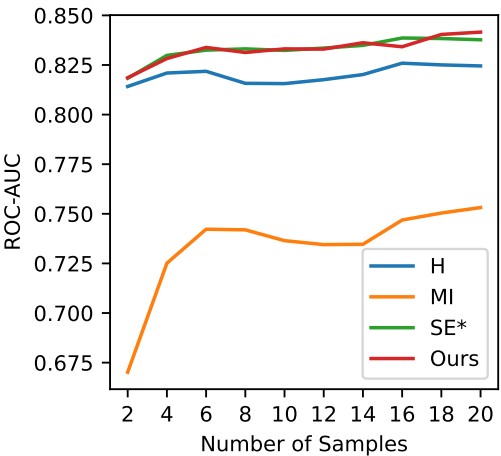

Figure 7: Change in ROC-AUC on TriviaQA of uncertainty quantification methods for Mistral-7B as we vary the number of output samples we produce from the LLM with multinomial sampling.

### E.6 Number of Samples Ablation

An ablation on the number of samples used in multinomial sampling is shown in Figure 7.

### E.7 ROUGE Threshold Ablation

An ablation on the choice of ROUGE threshold for evaluating Conceptual Uncertainty is shown in Figure 8. The threshold of 0.3 results in the highest performance, and we also use this threshold for the results throughout the paper.

### E.8 Answer Length Ablation

An ablation on how uncertainty metric performance varies with the average answer length of a question is shown in Figure 9.

### E.9 Confidence Calibration

While the uncertainty produced by our method, and similar approaches, represents an entropy, it can be hard to interpret in practice. We show that it is possible to post-hoc calibrate the uncertainty value produced by our approach into a confidence that the greedy answer is correct. We use Platt scaling (Platt et al., 1999), which trains a logistic regression classifier to map the uncertainty score for each question to a value in $[0, 1]$ representing the likelihood that the greedy answer is correct. This classifiers is trained on 20% of the evaluation set, or 80 samples, and we evaluate calibration on the rest of the test set. The calibration curve for our Conceptual Uncertainty method on TriviaQA using Mistral-7B is shown in Figure 10.

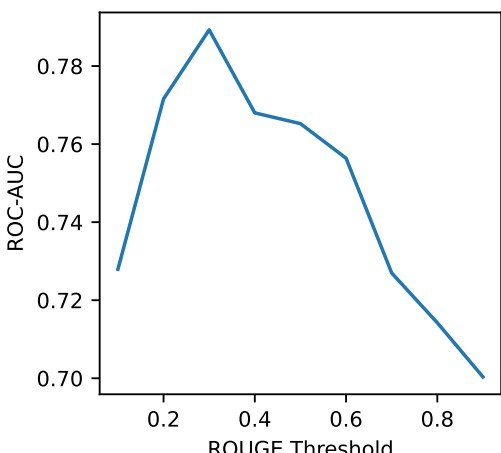

Figure 8: Ablation on the ROUGE threshold used for determining an answer's correctness. We show the ROC-AUC of Conceptual Uncertainty for the CoQA dataset with Mistral-7B as we vary the ROUGE threshold from 0.1 to 1.

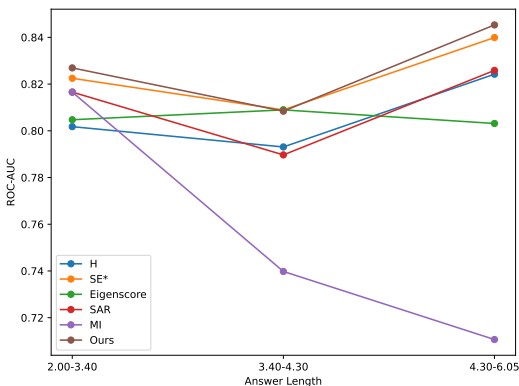

Figure 9: Change in ROC-AUC of uncertainty quantification methods on subsets of TriviaQA with equal size and model accuracy but different average response length using the Mistral 7B model.

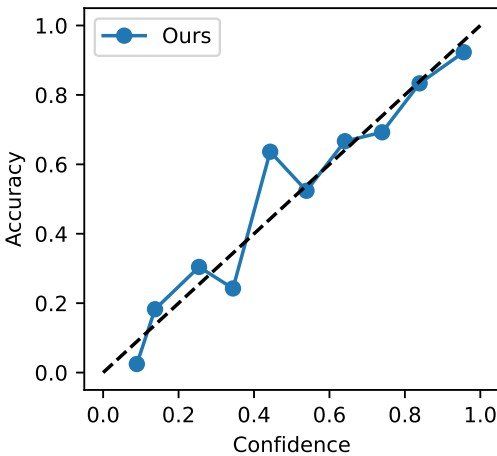

Figure 10: Calibration curve for our method on TriviaQA with Mistral-7B after Platt scaling.

Table 8: Existing LLM uncertainty methods as instances of our general framework. For the uncertainty metric, $\mathcal{H}(\{p_i\}_i) = -\sum_i \frac{p_i}{\sum_i p_i} \log \frac{p_i}{\sum_i p_i}$ is the regular entropy, $\hat{\mathcal{H}}(\{p_i\}_i) = -\sum_i p_i \log p_i$ is the unnormalized entropy, $\text{TokenSAR}(o_i) = (\Pi_k M(o_k|o_{k-1}\ldots q)^{w_k})^{1/K}$ where $w_k$ is the $k$th token importance derived using a sentence embedding model, $\text{SentSAR}(g_k) = -\log(p(g_k) + \frac{\sum_{j \neq k} w_{g_k,g_j} p(g_j)}{t})$ where $w_{g_k,g_j}$ is a sentence similarity derived using a sentence embedding model, and $\mathcal{I}(\{p(g_i)\}_i)$ is an estimate of the mutual information defined in (Abbasi-Yadkori et al., 2024).

| Method | Grouping | $p(g_i)$ | Metric |
|---|---|---|---|
| Entropy (Malinin & Gales, 2021) | None | $M(o_i|q)$ | $\mathcal{H}(\{p(g_i)\}_i)$ |
| SE (Kuhn et al., 2023) | NLI Entailment | $\sum_{o \in g_i} M(o|q)$ | $\hat{\mathcal{H}}(\{p(g_i)\}_i)$ |
| SE* (Aichberger et al., 2024) | NLI Entailment | $\sum_{o \in g_i} M(o|q)$ | $\mathcal{H}(\{p(g_i)\}_i) - \log \sum_{i=1}^{N} p(g_i)$ |
| Exact SE (Farquhar et al., 2024) | NLI Entailment | $\frac{|g_i|}{N}$ | $\mathcal{H}(\{p(g_i)\}_i)$ |
| SAR (Duan et al., 2024) | None | $\text{TokenSAR}(o_i)$ | $\frac{1}{K} \sum_{k=1}^{K} \text{SentSAR}(p(g_k))$ |
| MI (Abbasi-Yadkori et al., 2024) | Token F1 | $\frac{\sum_{o \in g_i} M(o|q)}{\sum_{i=1}^{N} M(o|q)}$ | $\mathcal{I}(\{p(g_i)\}_i)$ |
| P(True) (Kadavath et al., 2022) | Greedy only | $M(\text{True}|q, o_{\text{greedy}})$ | $-\log p(g_{\text{greedy}})$ |
| Eigenscore (Chen et al., 2024) | Last token cov. $\Sigma$ | $\frac{1}{|O|}$ | $-\frac{1}{|O|} \log \det(\Sigma + \alpha I_{|O|})$ |

# F  DESIGN SPACE OF EXISTING UNCERTAINTY MEASURES

## F.1  DESIGN SPACE ABLATION

Table 9: Design space exploration. H is the entropy, NLS is the $-\log \sum_i p(y_i|x)$ term, and ESE is the exact semantic entropy, $-\sum_c \bar{p}(c|x) \log \bar{p}(c|x)$.

| | TriviaQA | | TruthfulQA | | CoQA | | Avg. |
|---|---|---|---|---|---|---|---|
| Method | Raw | LN | Raw | LN | Raw | LN | |
| H | 0.837 | 0.801 | 0.543 | 0.457 | 0.716 | 0.682 | 0.673 |
| NLS | 0.853 | 0.803 | 0.711 | 0.459 | 0.668 | 0.675 | 0.695 |
| ESE | 0.788 | 0.793 | 0.600 | 0.510 | 0.669 | 0.656 | 0.670 |
| ESE + NLS | 0.863 | 0.837 | 0.733 | 0.491 | 0.717 | 0.698 | 0.723 |
| Ours | 0.858 | 0.840 | 0.749 | 0.580 | 0.650 | 0.736 | **0.736** |
| Avg. | **0.840** | 0.815 | **0.667** | 0.500 | 0.684 | **0.689** | |

