# OpenReview forum: "Measuring Language Model Uncertainty With Internal Concepts"
_ICLR.cc/2025/Conference — Submitted to ICLR 2025_

### Official Review · Reviewer_gA6K · 2024-10-25

**Soundness:** 2
**Presentation:** 3
**Contribution:** 2
**Rating:** 5
**Confidence:** 3

**Summary:**

This paper proposes to measure the predictive uncertainty of LLMs with concepts. This method defines semantics through the internal conceptual structure of the model rather than an external model, thereby assessing the correctness of open and closed questions.

**Strengths:**

The proposed method does not rely on external semantic models but instead measures uncertainty based on the internal conceptual structure of LLM itself. It avoids the bias of incorporating external model knowledge into LLM evaluations, making the uncertainty measurement more aligned.

Additionally, the method can adapt to both open-ended questions ("List a city in California") and closed questions ("Which state is number 31?”), which is very useful in many practical application scenarios.

Through experiments, the method shows an improvement, which demonstrates the effectiveness of conceptual uncertainty in hallucination detection.

**Weaknesses:**

The proposed method requires generating 10 to 20 candidate answers for each question to construct a conceptual structure, which has a high demand for computational or human annotation resources. This sampling amount will further increase, especially when dealing with open-ended questions.

Although the proposed method reduces reliance on external models, it still depends on labeled datasets (for example, accurate matching of correct answers to questions). This places high demands on the accuracy and coverage of the dataset being constructed. In some fields, obtaining sufficient labeled data may be quite difficult. I believe that this may limit the generalizability of this method.

The author used fixed rules and structures to construct concepts and categorized answer groups by defining implicational relationships. However, this may seem limited when dealing with more complex semantic relationships. I suggest the authors consider representation engineering [1] to enhance their concept discovery.

[1] https://arxiv.org/abs/2310.01405

**Questions:**

**Q1**: From the perspective of practical usage, how to balance the trade-off between computational cost and performance improvement?

**Q2**: Do you have plans to use self-supervised or representation techniques to automatically construct concepts in order to reduce annotation dependence?

---

> ### Author Response · Authors · 2024-11-18
>
> Thank you for your review. We want to emphasize that our method *does not require labeled datasets*, but rather computes an uncertainty value associated with a query, by sampling multiple model outputs and evaluating the output diversity. This approach is similar to prior works in that it requires generating multiple candidate answers, but we compute the output diversity over semantic objects which are defined using the model itself, and our notion of diversity takes into account semantic structure of outputs beyond simple partitions. We further emphasize that the concept structure is latent in the model’s probabilities, and is not rule-based.
>
> We respond in detail below.
>
> ### Detailed Responses:
>
> > *The proposed method requires generating 10 to 20 candidate answers for each question to construct a conceptual structure, which has a high demand for computational or human annotation resources. This sampling amount will further increase, especially when dealing with open-ended questions.*
>
> The requirement on sampling is a limitation of our work, as we mentioned on line 531 of the Discussion section. However, this limitation applies to *all sampling-based uncertainty methods* such as Semantic Entropy from Kuhn et al. In addition, the sampling requirement does not increase for open-ended questions, since we use 20 samples for both closed and open-ended questions. We also note that the sampling process is parallelizable, allowing us to sample all 20 responses at once given enough GPU resources.
>
> > *Although the proposed method reduces reliance on external models, it still depends on labeled datasets (for example, accurate matching of correct answers to questions).*
>
> To clarify, our method does not rely on labeled datasets, and labels are only used for evaluation. Both Figure 1 as well as Algorithm 1 where our approach is described contain no mention of labels.
>
> > *The author used fixed rules and structures to construct concepts and categorized answer groups by defining implicational relationships.*
>
> Our method is not rule-based and all structures are discovered from the LLM’s probability distribution.
>
> **Q1: Trade-off between computational cost and performance**
>
> We have included an ablation on the number of samples used compared to the resulting ROC-AUC of our method in Figure 6 of Appendix E.6. We find that for Mistral-7B on the TriviaQA dataset, using more samples results in minimal increase in performance, so we could use as few as 5 samples and still get almost the same performance as with 20 samples. When looking at prior work such as from Kuhn et al., we find that their comparable ablation study had similar results.
>
> **Q2: Reducing dependence on annotations**
>
> As clarified above, our method is already independent of any annotations.

---

> > ### Comment · Reviewer_gA6K · 2024-11-19
> >
> > I thank the comment from the authors. Based on this, I'll raise my score.

---

> > > ### Author Response · Authors · 2024-11-24
> > >
> > > Thank you for reviewing our response. As the discussion period ends soon, we would greatly appreciate it if you could share any remaining feedback or suggestions. Otherwise, if we have satisfactorily addressed your concerns, we hope that you would consider increasing your score to support our paper's acceptance.

---

> > > > ### Comment · Reviewer_gA6K · 2024-11-25
> > > >
> > > > I've already raised my score. After reading other reviewers' comments, I believe this is a reasonable rating.

---

### Official Review · Reviewer_hXUe · 2024-11-01

**Soundness:** 2
**Presentation:** 2
**Contribution:** 3
**Rating:** 5
**Confidence:** 4

**Summary:**

This paper evaluates the predictive uncertainty of LLMs for tasks such as QA. It assigns an uncertainty using a form of entropy that applies to semantic concepts. A key feature is the acknowledgement of hierarchy of concepts.  A high-level framework is proposed, followed by some specific approaches in the framework that group generated answers, apply some weights, and estimate uncertainty scores for answers using a weighted sort of entropy. Some experiments are performed, revealing good performance on AUROC for some datasets.

**Strengths:**

The authors propose a method that unifies some of the related work on semantic similarity and uncertainty quantification for LLMs. They propose some new ways to obtain clusters of answers and use those to compute uncertainty.

Another strength is that the results on some datasets seem promising. The choice of datasets is somewhat diverse.

**Weaknesses:**

In my view, the paper suffers from clarity related issues and seems too similar in many ways to previous work in the space. The paper goes out of its way to make the case that 1) the semantic understanding part of the work is internal rather than external, and 2) the work has theoretical basis as opposed to some prior work.

Re: 1): I could not fully understand the distinction between prior work on external models vs. the claim about using the internal model. Wouldn’t it be possible to use the same LLM for NLI, similar to Kuhn et al.? Why is this distinction important? I did not understand this from the paper, even though it came up several times.

Re: 2): the authors claim some theoretical basis but most of the discussion does not seem directly pertinent to the approach. The work feels like an extension of prior efforts. In my view, the claims of the paper are inconsistent with the presentation. The claims of theoretical justification either need to be softened or better explained.

As should be clear from many of my comments, I could not fully understand important details in the paper. Further details about my concerns and some associated questions are provided in the next section. Even though I’m familiar with much of the related work, I could not understand the new contributions. Perhaps the authors can clarify some aspects.

**Questions:**

Some questions and comments follow:

The abstract is too short. There are also some statements about % improvements that are unnecessary – I feel this is not suitable for situations where the datasets and baselines are not well known and standard.

In the second paragraph, the point about “external model” is too vague. This distinction b/w external and internal model was generally unclear to me and needs specification. I recommend adding some detail here. In general, there is a lot of repetition in the paper with details delayed to later, and when they do come later in Section 3, they are not clear enough for me.

The authors mention they focus on uncertainty (which is a function of a query) as opposed to confidence (which is a function of a query and generation). As I understand, the system computes a score U(q) for query q, for a model. How should one interpret this score? Does it represent the probability that at least one generation is correct for a query? Doesn’t this depend on the number of generations?

The title of Section 2.1 is too broad – there is a lot of literature that does not follow this paradigm.

Lines 127-129: The authors mention an exception but there are many types of exceptions given the vast literature on the subject of uncertainty quantification for LLMs. I recommend rewriting these lines.

What is the rationale behind the equation in line 215? Is it in the appendix and not in the main text?

Why is the NLI approach of Kuhn et al. for semantic relations between two answers not sufficient? Is that not a way to obtain I_q(a_i, a_j)?

There is an issue with the citation style in several places in the paper, like in line 242.

Why is there a min in equation 1?

What are the other strategies referenced in line 263?

It seems like conceptual entropy is mentioned in some older work. What was in termed in this work? Why was the old term not used?

Re: experiments: parameter tuning is done on 20% of what dataset? The full training dataset? And why is the test set so small, only 400 instances? Did the experiments repeat sub-samples for test set? Is the error mentioned anywhere?

Can the authors clearly explain how the AUROC was computed? This is not clear enough from line 327 and nearby places. When was a question deemed to be correct? When at least 1 answer is correct from the multiple answers generated?

How are correct questions determined when the datasets are combined? I don’t see this mentioned anywhere.

Did the authors run ablations on the Rouge-L threshold for determining correctness of queries for some datasets?

There are many more references in this space – please search for a review paper and add some of these if possible.

---

> ### Author Response · Authors · 2024-11-18
>
> Thank you for the review! We want to clarify that the distinction between internal concepts and external models is important for several reasons including 1) to uncover how LLMs intrinsically represent conceptual structures and uncertainty, and 2) to remove external dependencies and make uncertainty estimation possible in use cases or domains where a particular external model might not be appropriate.
>
> Regarding the differentiating factors compared to prior works, we note that: 1) We provide a framework for defining conceptual structure that supports semantic relations beyond simple partitions, unlike previous approaches. 2) We define an uncertainty over conceptual structures based on an uncertainty over truth-assignments, which offers several advantages, such as 2a) It provides uncertainty measures for open-ended questions, a capability not present in most prior work and 2b) It justifies the use of a highly effective normalization scheme, which was used without theoretical backing in previous studies. Overall our approach extends beyond the limitations of earlier work by being more theoretically grounded while also being more widely applicable.
>
> We provide detailed responses to your points below.
>
> ### Detailed Responses
>
> > *I could not fully understand the distinction between prior work on external models vs. the claim about using the internal model. Wouldn’t it be possible to use the same LLM for NLI, similar to Kuhn et al.? Why is this distinction important? I did not understand this from the paper, even though it came up several times.*
>
> We distinguish between using external models and internal information for uncertainty estimation to make clear that we measure uncertainty as an intrinsic state of a model. Since uncertainty is a property of a model, determining it through a form of introspection is practically relevant since we are interested in if the model internally reflects an awareness of its own lack of knowledge. As LLMs are used in highly diverse scenarios, dependencies on external task-specific models only limit the method’s applicability. In addition, modern LLMs already possess a wide range of knowledge that task-specific models often fail to capture. We show examples of the failures of an external model when evaluating LLM generations in Figure 3.
>
> As the reviewer noted, one could adapt existing techniques by prompting the LLM itself to perform the task of the external model. While this removes reliance on an external model, it will not address other limitations of existing approaches such as the inability to handle open-ended questions and not taking into account semantic structure other than partitions. Our method shows that it is possible to define internal concepts in such a way to address all these issues.
>
> To further support our point that simply replacing an external model with an LLM is not enough, we have performed a simple experiment where we replace the NLI model in Semantic Entropy with the prompt “Assuming that ‘<s1>’ is true, is the following entailed: ‘<s2>’? Output ‘Yes’ or ‘No’. Answer:”. The results for CoQA with Mistral-7B are shown below where we can see that replacing the external model (DeBERTa) with prompting of Mistral-7B actually decreases performance while our method improves over SE*:
>
> | Method | ROC-AUC |
> | --- | --- |
> | SE* (DeBERTa) | 0.799 |
> | SE* (Mistral-7B) | 0.752 |
> | Conceptual Uncertainty | 0.813 |

---

> ### Author Response · Authors · 2024-11-18
>
> > *the authors claim some theoretical basis but most of the discussion does not seem directly pertinent to the approach.*
>
> We have taken the feedback that the connection between Algorithm 1 (our practical implementation of Conceptual Uncertainty) and Section 3 was not clear and we added references to components of Algorithm 1 in Section 3.4 and modified the notation of Algorithm 1 to match the notation for Conceptual Entropy for disjoint groups derived in Section 3.3.
>
> Regarding the theoretical discussion not relating to the approach, we want to clarify that we provide theoretical support for the normalization scheme for entropy, which we find is highly important in practice and appears without justification in prior work. We also consider hierarchical semantic structures and provide an entropy-based uncertainty which takes such structure into account unlike prior work which is limited to partitions. For practical implementation, we derive how we can approximate the hierarchical structure as disjoint partitions allowing for efficient computation of the Conceptual Entropy. We note that the disjoint partitions take the hierarchical concept structure into account and that our computation of Conceptual Entropy over such partitions relies on its composition property over disjoint groups, which we derived.
>
> To further support the importance and relevance of Algorithm 1 to our full theoretical discussion, we have performed an additional experiment on TriviaQA for Mistral-7B by limiting to 6 candidate answers to make the computation tractable. The results are shown below:
>
> | Method | ROC-AUC | Time (s) |
> | --- | --- | --- |
> | Algorithm 1 | 0.826 | 0.496 |
> | Full Conceptual Uncertainty | 0.830 | 48.233 |
>
> Algorithm 1 takes under a second for computing the uncertainty on the TriviaQA test set while achieving almost the same ROC-AUC of the full approach. This experiment is included in Appendix E.5 of the revision.
>
>
> > *The work feels like an extension of prior efforts.*
>
> We want to clarify our contributions. First, we propose a method for estimating LLM uncertainty which works for both open and closed-ended scenarios and accounts for hierarchical semantic structure of candidate answers, unlike prior work. We show a method for extracting the necessary semantic structure directly from the LLM, resulting in a fully intrinsic uncertainty measure, while prior work mostly relies on external models to determine semantic structure. To compute the uncertainty over semantic objects, we present a new variant of the traditional entropy which is necessary to correct for the likelihood of the all-false truth assignment, and has connections to classic work from Renyi, as well as being highly important in practice.
>
> > *The abstract is too short. There are also some statements about % improvements that are unnecessary – I feel this is not suitable for situations where the datasets and baselines are not well known and standard.*
>
> We intended for the abstract to be short to provide readers with a brief overview of the work without overwhelming them with details. If the reviewer has any particular suggestions on what is missing, we will be happy to address them. In response to your feedback, we have revised the statement about percent improvement on the mixtures of datasets as well as clarified a few additional points. All changes to the abstract are denoted in blue in the revision.
>
> > *In the second paragraph, the point about “external model” is too vague.*
>
> By “external model” we mean a model different (different architecture, training data, or objective) from that whose uncertainty is being quantified. In the case of semantic entropy from Kuhn et al., the external model is a DeBERTa NLI model specifically trained for the task of evaluating semantic equivalence. If there was another point about the external model that was too vague, please let us know and we will be happy to address it.

---

> ### Author Response · Authors · 2024-11-18
>
> > *As I understand, the system computes a score U(q) for query q, for a model. How should one interpret this score? Does it represent the probability that at least one generation is correct for a query? Doesn’t this depend on the number of generations?*
>
> The score $U(q)$ from our approach, as well as most other entropy-based uncertainty methods, represents an entropy. In practice, the score itself doesn’t have an intuitive meaning, but comparisons between scores are meaningful (e.g. if $U(q_1) < U(q_2)$ then the model has less uncertainty on $q_1$ than $q_2$). Uncertainty is evaluated in this paper, as well as in prior works, for the task of predicting whether the greedy answer is correct. Since the ROC-AUC of the uncertainty is high for this task, the uncertainty can be calibrated using a monotonic mapping (or other post hoc calibration techniques) to an estimate of the probability that the greedy answer is correct. We have included a calibration curve for our method after applying post hoc calibration on TriviaQA in Figure 9 in Appendix E.9 of our revision. We show that our method can be successfully calibrated for easy interpretation if necessary.
>
> > *The title of Section 2.1 is too broad – there is a lot of literature that does not follow this paradigm.*
> > *Lines 127-129: The authors mention an exception but there are many types of exceptions given the vast literature on the subject of uncertainty quantification for LLMs. I recommend rewriting these lines.*
>
> We treat LLM uncertainty to be independent of the selected output (as discussed on line 119 of our original submission), following existing work, so we believe this is an accurate representation of the literature on LLM uncertainty. If the reviewer can point us to some specific examples of works which do not fit our framework, we will be happy to change the name of Section 2 to be more precise. Regarding the exception referenced on lines 127-129, we again are not aware of other types of exceptions.
>
> > *What is the rationale behind the equation in line 215? Is it in the appendix and not in the main text?*
>
> The rationale is that $I_q(a_i, a_j)$ should be 1 when the answer $a_i$ implies the answer $a_j$, and $I_q(a_i, a_j)$ should be less than one otherwise. Therefore, we use the likelihood ratio $\frac{M(a_j|qa_iq)}{M(a_j|q)}$ which can be interpreted as the pointwise mutual information between fact $qa_i$ and answer $a_j$ conditioned on the question $q$ to determine if adding the premise $qa_i$ into the context increases the likelihood of answer $a_j$. We clamp the likelihood ratio between zero and one since we want $I_q(a_i, a_j) \in [0, 1]$. We have included this rationale in our revision. We also note that the clamped ratio $\min(b/a, 1)$ is often used in fuzzy logic for fuzzy implication $a \Rightarrow b \in [0,1]$.
>
> > *Why is the NLI approach of Kuhn et al. for semantic relations between two answers not sufficient? Is that not a way to obtain I_q(a_i, a_j)?*
>
> The NLI approach from Kuhn et al. can be used to obtain $I_q(a_i, a_j)$. The problem with this approach is that we expect $I_q(a_i, a_j)$ to measure if answer $a_i$ *logically entails* answer $a_j$ for question $q$ with respect to the LLM’s internal knowledge, but the NLI task is textual entailment which slightly differs from logical entailment. For example, for the question “Name a city in California.”, the answer “San Francisco” does not textually entail the answer “Los Angeles” or vice versa for even the DebERTa-XLarge model, but it will be logically entailed for a reasonably capable LLM that knows that San Francisco and Los Angeles are in the same state.
>
> > *There is an issue with the citation style in several places in the paper, like in line 242.*
>
> We thank the reviewer for pointing the inconsistency out. We have fixed all citation style issues in the revision.
>
> > *Why is there a min in equation 1?*
>
> The $\min$ is used because a premise is only consistent with answers in S if $a$ implies *all* answers in $S$. The $\min$ is used to capture the desired *all or nothing* property of entailing a set of answers. We have added this intuition to line 254 of the revision.
>
> > *What are the other strategies referenced in line 263?*
>
> Some examples of other strategies are to directly verbalize the value of $I_q(a_i, a_j)$ from the model, or to learn $I_q(a_i, a_j)$ from data. We have added a short reference to these strategies in our revision, but this statement was meant as an invitation for future work rather than additional contributions of the paper.

---

> ### Author Response · Authors · 2024-11-18
>
> > *It seems like conceptual entropy is mentioned in some older work. What was in termed in this work? Why was the old term not used?*
>
> We mention that the conceptual entropy appears in older work from Renyi where it was called entropy for generalized distributions. We used the term “conceptual entropy” to distinguish this entropy from the entropy used in existing work on entropy-based uncertainty quantification. We also note that Renyi’s treatment did not consider distributions over a powerset algebra (all possible subsets of a set), which is a central part of our formulation of conceptual entropy.
>
> > *Re: experiments: parameter tuning is done on 20% of what dataset? The full training dataset? And why is the test set so small, only 400 instances? Did the experiments repeat sub-samples for test set? Is the error mentioned anywhere?*
>
> The parameter tuning was performed on 20% of the evaluation set, so 80 samples, and the test set was the remaining 320 samples. We used a total of 400 samples following prior work [1]. We have updated the Experiments section to mention these details following the reviewer’s feedback. Regarding the error, we included error bars for the TriviaQA dataset in Appendix E.2 of the original submission. We do not report error bars for all experiments due to computational requirements.
>
> [1]: Farquhar, S., Kossen, J., Kuhn, L., & Gal, Y. (2024). Detecting hallucinations in large language models using semantic entropy. Nature.
>
> > *Can the authors clearly explain how the AUROC was computed? This is not clear enough from line 327 and nearby places. When was a question deemed to be correct? When at least 1 answer is correct from the multiple answers generated?*
>
> To compute the ROC-AUC, we first assigned an uncertainty score to each question using each method and then assigned a binary correctness label to each question based on whether the greedy generation answer is deemed correct. Question correctness is determined using the different methods described on line 328-329 of the original submission. Given the uncertainty score for each question compared to the binary correctness label, we compute the ROC-AUC which is the area under the ROC curve (the true positive rate vs. the false positive rate for each threshold of the uncertainty metric). We have clarified these details in the revision.
>
> > *How are correct questions determined when the datasets are combined? I don’t see this mentioned anywhere.*
>
> Thank you for bringing this omission to our attention. When the datasets are combined, we measure question correctness for each individual question the same way that we would measure it in its original dataset. We have added this clarification to our revision in Section 4.2.
>
> > *Did the authors run ablations on the Rouge-L threshold for determining correctness of queries for some datasets?*
>
> In the paper, we used a ROUGE threshold of 0.3 since this was the value used in prior work. We have since performed an ablation on the ROUGE threshold and included the results in Figure 7 in Appendix E.7 where the threshold of 0.3 results in the best performance and increasing or decreasing the threshold decreases the performance of our method on CoQA.
>
> > *There are many more references in this space – please search for a review paper and add some of these if possible.*
>
> We have added the following references [1, 2, 3, 4] to our revision in response to your feedback. If you believe we have overlooked other relevant works, please let us know and we will gladly include them. Overall our related work section seems aligned with those in other recent papers in the field. One area that we did not focus on is classical uncertainty quantification, such as using Bayesian approaches, but these methods are not directly applicable to LLMs since they are computationally intractable for large models. Despite this, we have added a reference to a survey paper on such approaches [5].
>
> [1]: Manakul, P., Liusie, A., & Gales, M. J. (2023). Selfcheckgpt: Zero-resource black-box hallucination detection for generative large language models. EMNLP 2023.
>
> [2]: Chen, J., & Mueller, J. (2024, August). Quantifying uncertainty in answers from any language model and enhancing their trustworthiness. In Proceedings of the 62nd Annual Meeting of the Association for Computational Linguistics (Volume 1: Long Papers).
>
> [3]: Fomicheva, M., Sun, S., Yankovskaya, L., Blain, F., Guzmán, F., Fishel, M., ... & Specia, L. (2020). Unsupervised quality estimation for neural machine translation. Transactions of the Association for Computational Linguistics.
>
> [4]: Sun, M., Yan, W., Abbeel, P., & Mordatch, I. (2022). Quantifying uncertainty in foundation models via ensembles. In NeurIPS 2022 Workshop on Robustness in Sequence Modeling.
>
> [5]: Gawlikowski, J., Tassi, C. R. N., Ali, M., Lee, J., Humt, M., Feng, J., ... & Zhu, X. X. (2023). A survey of uncertainty in deep neural networks. Artificial Intelligence Review.

---

> > ### Comment · Reviewer_hXUe · 2024-11-26
> > **Thank you**
> >
> > Thanks to the authors for their detailed response. I found several clarifications useful, such as the internal/external model distinction. My fundamental opinion of the paper has not meaningfully changed as I appreciate both the strengths and weaknesses of the work.
> >
> > A comment about internal vs. external models -- if a particular model is better at a task like NLI, then I don't see why it should not be used for UQ for some other model in an approach that uses NLI. Also, re: my comment about the title of Section 2.1 being too broad -- I recommend looking at some review papers (like this one: https://aclanthology.org/2024.naacl-long.366/). I was merely emphasizing the breadth of the space.

---

> > > ### Author Response · Authors · 2024-12-04
> > >
> > > Thank you for your response.
> > >
> > > Regarding internal concepts vs. external models, as we discuss in our response above and in Section 2.2, our motivation for using internal concepts is twofold: 1) to measure uncertainty as an *intrinsic* property of the LLM and 2) to leverage an LLM’s knowledge of its own output. In our paper, we present a method to measure uncertainty using the structure of internal concepts, outperforming existing methods which rely on external NLI models. We do not argue that using internal concepts should perform better on a traditional NLI benchmark than a model trained for NLI, but we demonstrate that these NLI models provide less insight into an LLM’s uncertainty than using our internal concepts. In fact, our results in Table 3 show that methods such as SE and SE*, which rely on an NLI model, significantly underperform our method on the GSM8K dataset containing grade school math since the NLI model is not very useful in this domain. Finally, as stated in our original response, it is practically relevant to measure uncertainty without an external model since such a method will scale with the capabilities of the LLM and should be better suited to the model’s output distribution.
> > >
> > > Regarding the title of Section 2.1, we have looked at the referenced review paper, and our paper already includes references to all but one of the papers listed in their Table 2 showing “representative research on confidence estimation for LLMs” [1]. The one paper we did not reference is a method for measuring *confidence* in a particular answer rather than *uncertainty* of an LLM in the context of a question.
> > >
> > >
> > > [1]: Geng, J., Cai, F., Wang, Y., Koeppl, H., Nakov, P., & Gurevych, I. (2024, June). A Survey of Confidence Estimation and Calibration in Large Language Models. In Proceedings of the 2024 Conference of the North American Chapter of the Association for Computational Linguistics: Human Language Technologies (Volume 1: Long Papers) (pp. 6577-6595).

---

### Official Review · Reviewer_zr87 · 2024-11-03

**Soundness:** 2
**Presentation:** 2
**Contribution:** 3
**Rating:** 6
**Confidence:** 2

**Summary:**

The submission proposes a new approach to quantifying uncertainty of LLMs. The method, named Conceptual Uncertainty, performs a hierarchical grouping (via a lattice) of candidate answers on the basis of which is then derives a likelihood function, form which an entropy measure is derived. The resulting measure then provides a measure of uncertainty for the scenario.

The key novelty is the organisation of candidate answers in a lattice structure that captures more detailed relationships between answers than the partitioning into semantically equivalent answers generally considered by previous work. In particular, the lattice is induced by the partial order over answers that represents an answer being correct implying another answer being correct.

The method is then compared empirically to a range of methods from the literature on Mistral 7B, Falcon 7B, and Mamba 2.8B, with the proposed method of Conceptual Uncertainty performing best in most experiments.

**Strengths:**

S1) The combination of a likelihood function derived directly from the approximation of the partial ordering, as well as the variation on traditional entropy combine elegantly to an intuitively sensible measure of uncertainty.

S2) The concept lattice is, to the best of my knowledge, an original contribution to the study of LLM uncertainty and conceptually seems to be a step forward from considering only equivalence classes.

S3) In the empirical evaluation the method compares favourably against a wide range of methods from the literature. Notably experiments were performed with transformer-based and with state-space LLMs which is particularly relevant as the proposed approach is highly LLM dependent.

**Weaknesses:**

W1) To me, the presentation is confusing at times. While effort was made to provide ample examples, they seem under explained and often unclear. For example, as far as I understand the assignment likelihoods in Figure 1 seems unconnected to the lattice in the figure and overall the figure offers little insight at this point in the text without further explanation.

W2) The authors explain that a key distinction of their approach to mot approaches in the literature is that they derive their concept structure from the LLM. This is done by approximately recovering the partial order over concepts via a specific ratio of LLM output probabilities. This then naturally carries with it its own uncertainty based on the LLM which is ignored in this analysis. See also Q2.
Furthermore, I am not sure if it is not straightforward to adapt other methods to derive their respective concept structures (say semantic equivalence relations) via the LLM. This key distinction therefore seems somewhat weak to me.

**Questions:**

Q1) In order to estimate the uncertainty of answers for question q, the proposed method uses the ratio of LLM probabilities on question q itself to construct the lattice from which the uncertainty measure is then derived. This seems somewhat circular to me, and intuitively this approach would perform worse in situations with high uncertainty, as the approximated concept lattice would be less reliable. Could you clarify why this cyclic nature is not of greater concern?


Q2) The method relies critically on the approximation of the partial ordering of concepts by I_q. Have you performed any systematic comparisons of the approximate partial orderings to the real partial orderings?

---

> ### Author Response · Authors · 2024-11-18
>
> Thank you for the valuable review! We provide detailed responses to your points below.
>
> ### Detailed Responses
> > *… as far as I understand the assignment likelihoods in Figure 1 seems unconnected to the lattice in the figure and overall the figure offers little insight at this point in the text without further explanation.*
>
> Thank you for the helpful feedback regarding Figure 1. We have revised the caption of Figure 1 in our revision to clarify the connection between the truth assignment likelihoods in step 2 and the lattice in step 1. To clarify the point here, the structure from step 1 allows us to compute the truth assignment likelihood function. For instance, the likelihood of $(a=\top, b=\bot)$ for two answers $a$ and $b$ will be small depending on the concept structure (i.e. if $a\leq b$). If there are other parts of the figure that are unclear, please let us know and we will be happy to add more details.
>
> **Using external models vs. using the LLM itself**
>
> We want our method to be independent of an external model for two main reasons. First, using an external model is only beneficial when the external model possesses additional knowledge or capabilities compared to the model whose uncertainty is under study, but this is often not the case when evaluating LLMs. For instance, in the extreme case where the external model knows the correct answer to some questions, the uncertainty can entirely rely on the external model to classify if the output is correct or incorrect; however, this uncertainty will fail when the external model is itself uncertain. We therefore want to remove external dependencies so that our method can scale with LLMs. The second reason is that we believe that determining LLM uncertainty through a form of ‘introspection’ (looking within a model) is an interesting scientific question that also sheds light on the latent internal conceptual structures in LLMs.
>
> For determining LLM uncertainty through introspection, we could simply adapt existing techniques to use the LLM itself rather than an external model, but this would still have problems. For example, making the Semantic Entropy method from Kuhn et al. use the LLM itself for determining semantic equivalence will still fail for open-ended questions. Therefore, the problem is how to perform the introspection in a way which yields a useful uncertainty measure. In this paper, we show that we can do so by measuring uncertainty over our definition of internal concepts.
>
> **Q1: Circularity of concept lattice construction**
>
> We agree with the reviewer that the construction of a concept lattice from an LLM will have its own associated uncertainty. However, since determining if answer $a$ implies answer $b$ is much easier than determining if answer $a$ is correct in the first place, these types of uncertainty need not be correlated. For example, we may not know the answer to the question “When was Lincoln born?”, but we do know that if the answer is “February 12, 1809” then “February 12” is also correct. In addition, since we measure $I_q(a, b)$ as a likelihood ratio, we measure the influence of the premise $b$ on the likelihood of answer $a$ which removes the influence of the particular likelihoods of either $a$ or $b$.
>
>
> **Q2: Comparison with real implication orderings**
>
> Thank you for this question. We have performed an additional evaluation of the partial ordering from $I_q(a, b)$ using the SNLI dataset to show that the derived partial ordering aligns with an existing ordering. To evaluate $I_q(a, b)$, we use the binary classification task of determining the entailment direction of two sentences from SNLI. Since entailment is not symmetric, we can evaluate if $I_q(a, b)$ determines the correct ordering. The results are included below as well as in Figure 5 in Appendix E.4 of the revision where we include further experimental details.
>
> | Model | Accuracy |
> |-------|----------|
> | Random | 0.50 |
> | DeBERTa | 0.81 |
> | Joint Likelihood | 0.61 |
> | $I_q(a, b)$ | 0.68 |
>
> We see that $I_q(a, b)$ results in the correct partial ordering of sentences 68% of the time which is significantly better than the random baseline as well as the joint likelihood baseline from [1] where they determine the partial order using the likelihood of the string “a_i a_j” compared to “a_j a_i”. While these methods are not as good at this task as the DeBERTa NLI model, a supervised model trained specifically for the NLI task, we only use the partial ordering as a component of our uncertainty method rather than as the final output.
>
> [1]: Liu, T. Y., Trager, M., Achille, A., Perera, P., Zancato, L., & Soatto, S. (2023). Meaning representations from trajectories in autoregressive models. ICLR 2024.

---

> > ### Comment · Reviewer_zr87 · 2024-11-20
> >
> > Thank you for the extensive response.
> >
> > I can sympathise with the challenges regarding the circularity of the uncertainty in these problems. However, I do think that these are a critical matter to consider in the future study of such "internal" uncertainty measures and therefore  would ideally warrant a more careful and thorough discussion in the paper.
> >
> > With respect to the experiments for Q2 I am somewhat confused. From reading Section E.4 and the random baseline being 0.5 I believe to understand that this evaluation only concerns itself with partial orders of 2 elements? That seems to me as insufficient to make the desired point, and in a way more confusing than having no numbers at all as it requires careful reading to notice that this is a special case, somewhat detached from the general point of the paper (as the lattice structure becomes essentially irrelevant in the tested setting with 2 elements). Could you please clarify on why you chose to address Q2 in this limited way?
> >
> > I assume there is some difficulty in automated generation of ground truth orderings in general, but it would already be insightful to see how this approximate partial ordering compares to manually constructed ground truth on some selected sampels (with more than 2 elements).

---

> ### Author Response · Authors · 2024-11-24
>
> Thank you for the response.
>
> We have added a discussion regarding the concern of circularity when measuring uncertainty using internal concepts to the Discussion section (lines 530-534) of the revision. We emphasize that uncertainty occurs at different levels: the conceptual structure captures what is (simultaneously) *possible*, whereas output correctness records what is actually *true*. This distinction hopefully illustrates why circularity is avoided and why extracting conceptual structures is more robust and useful.
>
> Regarding the experiments for Q2, we note that a partial order is in fact a binary relation on a set ($R(a,b)\ \text{iff}\ a \le b$). Evaluating the relationship between two elements is not a special case of a general partial ordering since the binary relation is the core of a partial order. Our experiment aims to measure whether our method can successfully detect logical implication between individual pairs of inputs; if this is possible, it can be applied to infer logical implications across larger sets of answers.
>
> For insight on the lattice structure recovered by our method, Figure 2 shows the lattice for a set of answers for an example question where we show the joint likelihood (derived from $I_q(a, b)$) aligns closely with the expected structure. We also updated Appendix E.4 with three additional qualitative examples of the ordering derived from our method on simple sentences constructed from WordNet. In Figure 6 we compare the ordering from our method to the true ordering from WordNet, and we show our ordering mostly aligns with the ground truth.

---

> > ### Comment · Reviewer_zr87 · 2024-11-25
> >
> > I would like to thank the authors for all the additions and the interesting discussion.
> >
> > Two elements is a special case because there are only 4 trivial partial orders (the two total orders, a b incomparable, and a=b), and all depend only on the prediction for a single binary relationship. The challenges around using an approximation of the orderings via I_q intuitively seem more present when it is used to reconstruct a complex lattice, as it depends on getting all/most of the binary relationships right. Indeed, 2 elements is the only case where guessing I_q cannot yield a structure that is not a valid partial ordering. For more elements, the approximation can for example yield cases where antisymmetry is violated. While I understand that the authors are primarily concerned with practice, I am still concerned about the lack of insight into the many complex challenges that come with approximating it, given that the lattice structure is so core to the proposed approach.
> >
> > The joint likelihoods of Figure 2 are not enough to reconstruct I_q(a,b) and I would have been curious to see how the lattice induced by I_q matches up with the expected structure.
> >
> > Given the discussion and the comments of the other reviewers I believe the rating as it stands is appropriate.

---

> > > ### Author Response · Authors · 2024-11-26
> > >
> > > In case the reviewer missed this in our last reply (we may not have emphasized it enough), we would like to reiterate that, based on the reviewer’s feedback, we conducted an additional qualitative experiment to visualize the orderings derived from our method. As shown in Appendix E.4, Figure 6, the results show that our derived ordering mostly aligns with the ground truth.
> > >
> > > While we completely agree that exploring the robustness and accuracy of these latent structures would be very interesting—for instance, investigating the extent to which transitivity is respected—we believe that this should be a separate project, and that for the purpose of estimating uncertainty, simply extracting noisy structures is appropriate. This is consistent with prior works, which have employed external NLI models for semantic grouping without studying the robustness of such structures (e.g., examining the extent to which pairwise semantic equivalence satisfies the properties of an equivalence relation). We also note that we make use of *fuzzy* orderings, which means that the lack of an exact order structure is not a concern, and that any relation $I_q$ would give rise to a valid truth-assignment likelihood function.
> > >
> > > Thank you again for your thoughtful feedback and for engaging in this discussion! We appreciate your support for the acceptance of our paper.

---

### Official Review · Reviewer_pZNk · 2024-11-04

**Soundness:** 3
**Presentation:** 3
**Contribution:** 3
**Rating:** 6
**Confidence:** 2

**Summary:**

To address limitations in existing uncertainty evaluation metrics—such as dependence on external models lacking equivalent knowledge to the evaluated LLM, inability to handle open-ended questions, and failure to leverage hierarchical semantic structures for more effective grouping—the authors propose a framework called Conceptual Uncertainty for assessing LLM uncertainty. Specifically, they introduce an internal concept structure to define the distribution over possible truth assignments to answers, which is then used to calculate the newly-introduced conceptual entropy as a measure of uncertainty. Extensive experiments demonstrate that this proposed metric outperforms baseline methods in handling both open-ended questions and those questions with long answers.

**Strengths:**

- A novel approach that effectively leverages the inherent hierarchical semantic structures and the internal knowledge of the evaluated LLM to improve the precision of uncertainty evaluation.
- A clear presentation of the motivation and key challenges addressed by the framework.
- Intuitive figures and examples that aid readers in understanding the newly introduced concepts.
- Experiments conducted on both challenging cases, covering both closed- and open-ended questions, as well as questions with short and long answers.

**Weaknesses:**

- The clarity of the paper could be further enhanced, as noted in the questions below.
- There appears to be an inconsistency between the algorithm and the main text; specifically, lines 5 and 6 in Algorithm 1 seem more aligned with standard uncertainty evaluation practices rather than fully reflecting the conceptual entropy measure introduced in the paper. If the algorithm is correct, it would be helpful to clarify the connections between the functions in the algorithm and the functions in the main text.

**Questions:**

- In line 67, Why can the connection between a and b be quantified by the likelihood ratio of strings of the form qbqa and qa? A more intuitive explanation would be helpful.
- Why does Equation (1) hold? Could you provide a more intuitive rationale for it?
- Some notations in Equation (2) are not formally defined, which may lead to reader confusion.
- How the expected (internal) concepts are determined?
- In line 319, you mention that 20% of the samples are used for parameter tuning. Which specific hyperparameters are tuned?
- Have you evaluated the performance of the new metric on larger-scale LLMs, such as GPT-4 or others?
- How many sample answers are generated to evaluate conceptual uncertainty for each question? How does the sample size impact the performance of the conceptual uncertainty metric?
- In line 360, why is the question “What is my favorite integer between X and Y, inclusive?” considered a closed question?

---

> ### Author Response · Authors · 2024-11-18
>
> Thank you for your detailed review! We used your feedback to improve several parts of the paper by providing additional intuition for equations and clarifying some of the discussions. We improved the presentation of our practical approach in Algorithm 1 by aligning notation with the main text as well as adding more references to the algorithm from the text. We also included an evaluation of the complete method presented in the paper by limiting to 6 output samples to make the method tractable, and we show that Algorithm 1 is significantly more efficient to compute while resulting in similar performance. This additional experiment is included below and in Appendix E.5.
>
> We respond to your points in detail below.
>
> ### Detailed Responses
> > *There appears to be an inconsistency between the algorithm and the main text; specifically, lines 5 and 6 in Algorithm 1 seem more aligned with standard uncertainty evaluation practices rather than fully reflecting the conceptual entropy measure introduced in the paper. If the algorithm is correct, it would be helpful to clarify the connections between the functions in the algorithm and the functions in the main text.*
>
> Thank you for the feedback on this point. The algorithm was correct, but it used different notation from the main text. In response to your feedback, we have updated Algorithm 1 to be consistent with the notation from the end of Section 3.3 for computing the Conceptual Entropy. We also added explicit references to the Algorithm in Section 3.4 to better explain the connection to the main text. To be clear, the algorithm implements a practical version of the Conceptual Uncertainty by using the weighted composition property of Conceptual Entropy over disjoint groups. As explained in Section 3.4, this is necessary due to its computational complexity.
>
> We also performed an additional evaluation comparing Algorithm 1 and the complete Conceptual Uncertainty method on TriviaQA for Mistral-7B by limiting to 6 candidate answers to make the computation tractable. The results are shown below:
>
> | Method | ROC-AUC | Time (s) |
> | --- | --- | --- |
> | Algorithm 1 | 0.826 | 0.496 |
> | Full Conceptual Uncertainty | 0.830 | 48.233 |
>
> Algorithm 1 takes under a second for computing the uncertainty on the TriviaQA test set while achieving almost the same ROC-AUC of the full approach. This experiment is included in Appendix E.5 of the revision.
>
> > *In line 67, Why can the connection between a and b be quantified by the likelihood ratio of strings of the form qbqa and qa? A more intuitive explanation would be helpful.*
>
> The intuition for the likelihood ratio of strings of the form $qbqa$ and $qa$ is that it can be interpreted as a pointwise mutual-information between $qb$ and $a$ conditioned on question $q$. When the likelihood ratio is large, this means the presence of $qb$ in the prompt increases the likelihood of $a$ as a response to $q$. We have added this intuition to lines 78-79 in the revision.
>
> > *Why does Equation (1) hold? Could you provide a more intuitive rationale for it?*
>
> We provided some intuition in line 254 of the original submission, but we have further clarified the equation in lines 254-255 of the revision. First, for a given premise $a$, the equation $\min_{a’\in S} I_q(a, a’)$ quantifies the likelihood that premise $a$ is consistent with all the answers in $S$. The $\min$ is used because a premise is only consistent with answers in S if $a$ implies *all* answers in $S$. The full Equation 1 is then an expected value over premises of the likelihood that all answers in S are true.
>
> > *Some notations in Equation (2) are not formally defined, which may lead to reader confusion.*
>
> We believe the notation in Equation 2 is defined, but to be clear, we added a clarification to the text that $\tau = (\tau_1, \dots, \tau_n)$. If the reviewer still believes there is something that lacks a formal definition, we will be happy to fix the problem.
>
> > *How the expected (internal) concepts are determined?*
>
> We believe that by “expected concepts” you are referring to the highlighted concepts in Figure 2. These concepts are determined by our intuition of what the concept structure should look like since intuitively Bob playing piano or violin implies that he plays music, and Bob playing tennis implies he plays sports. We then show that our intuition actually aligns well with our concept extraction method, but this is a qualitative observation.
>
> > *In line 319, you mention that 20% of the samples are used for parameter tuning. Which specific hyperparameters are tuned?*
>
> The parameters that are tuned are the distance threshold $\epsilon$ in Algorithm 1, and the use of length normalized likelihoods versus raw likelihoods. We also tune the use of length-normalized likelihoods for existing approaches. We have clarified this on lines 315-316 in the revision.

---

> ### Author Response · Authors · 2024-11-18
>
> > *Have you evaluated the performance of the new metric on larger-scale LLMs, such as GPT-4 or others?*
>
> Our evaluation is limited to models of at most 7B parameters due to resource constraints, but we would love to perform an evaluation on larger models in the future.
>
> > *How many sample answers are generated to evaluate conceptual uncertainty for each question? How does the sample size impact the performance of the conceptual uncertainty metric?*
>
> Thank you for pointing out this unintentional omission. For all experiments, we sampled 20 answers using multinomial sampling. We have added this to line 340 of the revision. We have also performed an ablation on the impact of sample size on performance for TriviaQA and included it in Figure 6 in Appendix E of the revision. Similar to existing work, we find that sample size has minimal impact on performance for our method above 5 samples.
>
> > *In line 360, why is the question “What is my favorite integer between X and Y, inclusive?” considered a closed question?*
>
> We consider this a closed question because there can only presumably be one correct answer. For example, if I say “What is my favorite number between 1 and 10” then the assumption is that I have a favorite number in this range and you must guess what it is.

---

> > ### Comment · Reviewer_pZNk · 2024-11-25
> >
> > Thank you for the detailed responses. The revisions and additional clarifications are helpful. I would like to keep my score as "above the acceptance threshold".

---

### Official Review · Reviewer_3TEp · 2024-11-04

**Soundness:** 3
**Presentation:** 4
**Contribution:** 3
**Rating:** 6
**Confidence:** 3

**Summary:**

The paper is well written and structured.
It deals with a way to evaluate the uncertainty of a Large Language Model’s (LLM)  in terms of output meaning as corresponding the questions submitted to it.
The authors approach relies on “internal concepts ” i.e. semantic objects from the LLM itself and not derived from an external semantic model of the language thus avoiding introducing differing/contradicting semantics between the external model and the LLM.
Their approach is entropy-based.

**Strengths:**

Quantified uncertainty is meant to evaluate the LLM’s responses and thus can help detect highly probable (semantically) odd responses (LLM hallucinations).
The approach tackles both open and close-ended queries and through what is termed “conceptual uncertainty”. It applies a query-dependent “truth-assignment likelihood” function to estimate LLM answers “truthfulness”.
Based on the various LLM responses’ “truthfulness” values, an entropy-based  uncertainty measure is computed. The entropy measure introduced is called “Conceptual Entropy”, only relies on non-all-false LLM responses truth  assignments, and accounts for mutually exclusive concepts and output length differences through length-normalized likelihoods.
Finally uncertainty is computed as a function of answers semantic groups (of model outputs) and their likelihoods.

**Weaknesses:**

See questions' section

**Questions:**

S: Statement. Q: Question.

S1: In “2 A GENERAL FRAMEWORK FOR LLM UNCERTAINTY”, you state “ We use the term uncertainty as opposed to confidence as we view uncertainty as independent of the selected output to the query”.
Q1.1: Can you further clarify ?

S2: In “COMPONENTS OF LLM UNCERTAINTY”, you state “We further decompose existing diversity measures into three key components”.
Q2.1: Are you referring to “2) measuring some form of answer diversity” form of uncertainty mentioned earlier in the paragraph?
Q2.2: Isn’t this in contradiction with  your statement in Q1? Or
Q2.3 Do you consider that diversity of answers is independent from selected output?

S3: In “3.1 INTERNAL CONCEPTS”, You state “Given a set of candidate answers A = {a1, . . . , an} ⊂ A∗ for a query q, we view the corresponding “concept structure” as determined by a partial ordering where ai ≤ aj indicates that if ai is a valid output for q, then aj is also a valid output.”
	Q3.1: This is not clear to me. Doesn’t this make all model candidate answers, whatever their place in the order, valid as a hypothesis? No order depth limit as deeper positions might mean very uncertain?

S4: In “4.1 SETUP-Metric”, you state “randomly selected correct question having a larger uncertainty score than a randomly selected incorrect question.”.
	Q4.1 can you clarify “correct/incorrect question?”

---

> ### Author Response · Authors · 2024-11-18
>
> Thank you for reading our paper and providing valuable feedback! We respond to your specific questions below, but please let us know if there are any additional or remaining major concerns.
>
> ### Detailed Responses
> **Q1.1: Clarification between confidence and uncertainty**
>
> When prompting an LLM with a question, an answer is generated based on the sampling technique and is presented to the user. A confidence measure takes the generated response into account and produces a score reflecting how likely the particular response is correct. On the other hand, an uncertainty measure cannot leverage the information of the particular answer that was output. Therefore, the uncertainty is a property of the question and reflects the likelihood that the LLM will respond with a correct answer, rather than the likelihood that one answer is correct. In practice, the uncertainty can be estimated by considering the diversity of outputs.
>
> **Q2.1, Q2.2, and Q2.3: Clarifications on LLM uncertainty decomposition**
>
> To clarify, the three key components refer to components of the answer diversity measure. We believe the confusion is between the terms “diversity measure” and “uncertainty measure”. In the paper, we state “We further decompose existing *diversity measures*…”, but to make this clearer we explicitly reference “step 2” in the revision. This is not in contradiction to our method being independent of the selected output since output diversity does not depend on which candidate answer is chosen.
>
> **Q3.1: Partial ordering question**
>
> We agree with the reviewer that any candidate answer is a valid answer since the model can plausibly output it when using multinomial sampling. In our approach, answers are representatives of a hypothesis, but the hypotheses we consider are actually all complete *truth assignments* to all answers. Since we quantify the uncertainty over the distribution over such hypotheses, the position of particular answers in the partial order is not directly used. We hope future work can further study such hierarchical concept structures and their relation with answer confidence, as mentioned by the reviewer.
>
>
> **Q4.1: Question correctness for ROC-AUC**
>
> We use the terms “correct question” and “incorrect question” to mean a question which the model answers correctly via its greedy generation and one answered incorrectly via the greedy generation respectively. This is clarified in the sentence starting on line 327 in the original submission. If there is something else that is unclear about this, we will be happy to address it.

---

> > ### Comment · Reviewer_3TEp · 2024-11-20
> > **After author responses**
> >
> > Thanks to the authors for their responses.
> > Based on those, I raised my rating to "Marginally above the acceptance threshold".

---

### Author Response · Authors · 2024-11-18
**General Response**

We thank all reviewers for their valuable and constructive feedback. In response to the comments received, we have made some revisions to the paper, with the key changes highlighted in blue.

The main modifications include:
- A revised abstract.
- Clarified equations and notations where requested.
- Additional experiments and ablations included in Appendix E.4-9.

We address the feedback from individual reviewers in detail in the responses below.

---

### Meta-Review · Area_Chair_fKzB · 2024-12-21

**Metareview:**

This paper introduces a framework named Conceptual Uncertainty for evaluating LLM uncertainty, aiming to overcome the shortcomings of existing metrics. It employs an internal concept structure to define distributions over possible truth assignments for answers, utilizing this structure to compute conceptual entropy as an uncertainty measure. Extensive experiments suggest that this metric surpasses traditional methods in addressing both open-ended questions and queries requiring lengthy responses. Despite these contributions, the paper appears to be somewhat marginal, lacking a strong, distinctive stance. My assessment aligns with the reviewers' concerns regarding its novelty and overall impact. The proposed framework bears resemblance to prior studies, and while the authors mention a theoretical underpinning, it is tangentially related to their methodology. Additionally, the distinction between internal and external model components requires further clarification. Several reviewers also raised concerns on the presentation of this paper and necessary connections to improve the logics.

Given these issues—particularly the questions about its originality and significance—I concur that the paper may not yet meet the publication standards for ICLR.

**Additional Comments On Reviewer Discussion:**

This is a rather borderline paper with no strong opinions. Based on my reading, I think the reviewers are right on some remaining concerns regarding the novelty and contribution. The paper is similar in many ways to previous work in the space. The authors claim some theoretical basis but most of the discussion does not seem directly pertinent to the approach. And more clarification on internal/external model distinction is needed to justify the significance of this work. Several reviewers also raised concerns on the presentation of this paper and necessary connections to improve the logics.

In any case, this is clearly a borderline paper. It is interesting but also has a low originality and weak significance. For that reason I think it is not ready for ICLR.

---

### Decision · Program_Chairs · 2025-01-22

Reject